# Combining Double-Dose and High-Dose Pulsed Dapsone Combination Therapy for Chronic Lyme Disease/Post-Treatment Lyme Disease Syndrome and Co-Infections, Including Bartonella: A Report of 3 Cases and a Literature Review

**DOI:** 10.3390/microorganisms12050909

**Published:** 2024-04-30

**Authors:** Richard I. Horowitz, John Fallon, Phyllis R. Freeman

**Affiliations:** 1New York State Department of Health Tick-Borne Working Group, Albany, NY 12224, USA; 2Hudson Valley Healing Arts Center, Hyde Park, NY 12538, USA; johnf@hvhac.com (J.F.); research@hvhac.com (P.R.F.)

**Keywords:** Lyme disease, post-treatment Lyme disease syndrome (PTLDS), dapsone combination therapy (DDSCT), double-dose dapsone combination therapy (DDDCT), high-dose dapsone combination therapy (HDDCT), babesiosis, bartonellosis, fluorescent in situ hybridization (FISH), persistent infection, COVID-19

## Abstract

Three patients with relapsing and remitting borreliosis, babesiosis, and bartonellosis, despite extended anti-infective therapy, were prescribed double-dose dapsone combination therapy (DDDCT) for 8 weeks, followed by one or several two-week courses of pulsed high-dose dapsone combination therapy (HDDCT). We discuss these patients’ cases to illustrate three important variables required for long-term remission. First, diagnosing and treating active co-infections, including *Babesia* and *Bartonella* were important. *Babesia* required rotations of multiple anti-malarial drug combinations and herbal therapies, and *Bartonella* required one or several 6-day HDDCT pulses to achieve clinical remission. Second, all prior oral, intramuscular (IM), and/or intravenous (IV) antibiotics used for chronic Lyme disease (CLD)/post-treatment Lyme disease syndrome (PTLDS), irrespective of the length of administration, were inferior in efficacy to short-term pulsed biofilm/persister drug combination therapy i.e., dapsone, rifampin, methylene blue, and pyrazinamide, which improved resistant fatigue, pain, headaches, insomnia, and neuropsychiatric symptoms. Lastly, addressing multiple factors on the 16-point multiple systemic infectious disease syndrome (MSIDS) model was important in achieving remission. In conclusion, DDDCT with one or several 6–7-day pulses of HDDCT, while addressing abnormalities on the 16-point MSIDS map, could represent a novel effective clinical and anti-infective strategy in CLD/PTLDS and associated co-infections including *Bartonella*.

## 1. Introduction

Lyme disease, a bacterial infection caused by *Borrelia sensu lato* species, is increasing worldwide, with a prevalence rate estimated at 14.5% of the global population [1]. The recent implementation of a revised case definition for Lyme disease by the Centers for Disease Control and Prevention (CDC) reported that case counts are also rising in the United States (US), where the incidence was 1.7 times the annual U.S. average in 2017–2019, an overall 68.5% increase, rising with patient age in 2022 [2]. Finding effective solutions for Lyme disease is paramount as the infection can result in a debilitating chronic disease if not diagnosed and treated promptly early in the course of the illness [3]. Despite early treatment, a significant proportion of individuals, estimated between 10 and 35% [4], can go on to develop post-treatment Lyme disease syndrome (PTLDS) in the United States (US) [5], an incapacitating chronic fatiguing, musculoskeletal, cardiovascular, and neuropsychiatric illness [6]. Approximately 21% of Belgian European patients exposed to Lyme borreliosis (LB) and treated for disseminated Lyme disease will also develop PTLDS [7], although exact numbers of PTLDS cases in Europe are unavailable due to lack of a widely adopted standardized case definition for the disease [8]. In one small US study, among those diagnosed with PTLDS, 6 months after the initial diagnosis, 20% still complained about widespread pain, 36% reported new onset of fatigue, and 45% reported ongoing neurocognitive difficulties, which significantly affected the quality of their lives [9]. In a larger systemic review, including 46 studies, a poorer quality of life was confirmed in those suffering from PTLDS, with sequelae including persistent muscle and joint pain, nerve pain (paresthesia’s), insomnia, poor appetite, and cognitive difficulties [10]. The annual aggregate economic burden of CLD/PTLDS in the US is estimated between $345 and 968 million (2016 US dollars) [11], and in Europe (Belgium), direct and indirect costs have been estimated at €19.4 and 151.5 million, respectively [12]. These numbers are likely underestimating the significant societal impact of the disease since EM rashes may not be the initial sign or symptom of early Lyme disease [13], with some studies reporting rates of EM rashes in the CLD/PTLDS population as low as 20% [14]. Standard laboratory testing may also miss the expanding range of *Borrelia* species, now associated with acute and chronic illness, where the lack of a gold standard for diagnosing the broad range of *Borrelia* species makes it difficult to produce accurate statistics [15].

To date, an estimated two million individuals in the US have been diagnosed with PTLDS [16], but due to overlapping symptoms with myalgic encephalomyelitis (ME), fibromyalgia (FM), and long COVID [17], which collectively affects an estimated 14% of the United States [18,19,20] and up to 49% of patients in Europe [19,21,22], the prevalence of PTLDS may be much higher. Varying case definitions, heterogeneous study design, duration of follow-up, and differing methods of measurement make it difficult to determine the exact magnitude of the problem [22]. Jason et al. (2022) determined that two years after having a COVID infection, 55% still had one or more symptoms [23], and recently, a positive correlation between COVID-19 severity and previous exposure of patients to *Borrelia* species (spp.) has been reported in the literature [24]. This needs to be confirmed in large clinical studies, especially since *Borrelia* has been spreading rapidly, with an estimated global seroprevalence of 14.5% [1], while a full understanding of the pathogenesis of ME, FM, long COVID, and chronic Lyme disease (CLD)/PTLDS is still emerging [25,26,27,28,29].

A multifactorial medical model, used to help diagnose and treat overlapping factors associated with CLD/PTLDS, known as multiple systemic infectious disease syndrome (MSIDS), correctly predicted 16/16 factors associated with long COVID, several years before the emergence of the coronavirus pandemic [30]. Both diseases share similar pathogenic mechanisms, where respective infections result in inflammation and immune dysfunction [31,32] with potential endothelial dysfunction [33,34]. The 16-point MSIDS map consists of six possible overlapping sources of inflammation that increase clinical symptomatology, including acute/chronic bacterial, viral, parasitic (e.g., *Babesia*), and/or fungal infections [30] with or without the persistence of bacterial fragments (peptidoglycans) [35]; abnormalities in the microbiome of the gastrointestinal (GI) tract; intestinal hyperpermeability with food sensitivities/mast cell activation; environmental toxicity (e.g., mold, heavy metals, and other environmental toxins); mineral and vitamin deficiencies; and sleep disorders/insomnia [36,37]. Downstream effects of one or multiple sources of inflammation on the MSIDS map, due to excessive free radical/oxidative stress, include potential immune dysregulation with immune deficiency and/or autoimmunity, autonomic nervous system dysregulation with postural orthostatic tachycardia syndrome (POTS) and dysautonomia, hormonal dysregulation (low adrenal function, low sex hormones, and thyroid abnormalities), mitochondrial dysfunction, liver function abnormalities, resistant pain syndromes, as well as neuropsychiatric (mood) and central nervous system symptoms, including memory/concentration problems [30]. Any or all 16 factors on the MSIDS map have been shown to be present in varying degrees in those suffering from CLD/PTLDS [38,39]. In long COVID, we see similar overlapping causes of inflammation and downstream effects with viral persistence and viral reactivation [40], the persistence of viral fragments [41,42], superimposed bacterial infections like Lyme disease [24,43], microbiome abnormalities [44], intestinal hyperpermeability with *Candida* [45] with or without mast cell activation [46], sleep disorders [47], environmental toxin exposure (air pollution) [48], nutritional deficiencies [49], immune dysfunction [50], autoimmunity [51], endothelial damage with micro clots [52], POTS/dysautonomia [53], hormonal dysregulation (adrenal dysfunction) [54], malfunctioning mitochondria [55], liver injury [56], chronic pain syndromes [57], neuropsychiatric disorders [58] as well as chronic cognitive difficulties with memory/concentration problems [59]. These have all been associated with long COVID, i.e., infection-associated chronic illness. One primary difference in the overlapping MSIDS variables between the two diseases, namely CLD/PTLDS and long COVID, apart from the infectious source driving the inflammation and immune dysfunction, is that the endothelial dysfunction seen in long COVID results in symptomatic micro-clots more often [60], whereas in CLD, abnormal coagulation parameters do not usually result in symptomatic disease states [61]. Infection with *Borrelia miyamotoi* and/or relapsing fever *Borrelia* are the exceptions as they have been reported to cause life-threatening hematologic complications [62].

*Bartonella* infections also may cause overlapping symptoms seen in chronic Lyme disease/PTLDS [63], apart from co-infections with *Babesia* spp. [64], and *Bartonella*’s frequency as an important Lyme disease-associated co-infection is being reported more frequently in the medical literature [38,39]. Co-infections including *Babesia* spp. and *Bartonella* spp. are known to potentially increase the severity of other tick-borne disorders like Lyme disease [65,66]. As climate change increases global temperatures, resulting in rising numbers of vector-borne infections, including *Borrelia burgdorferi* and *Bartonella* [67], due to escalating populations of ticks, fleas, lice, and mosquitos, previously controlled vector-borne diseases are now resurging or resurfacing in new geographic locations, including newly identified pathogens, adversely impacting the prevalence of chronic illnesses [68]. One of the markers of an active infection with *Bartonella*, i.e., elevated levels of vascular endothelial growth factor (VEGF) [69], has recently been found not only in those with CLD/PTLDS but also in those suffering from long COVID [70], implying a potential overlapping role in driving chronic symptomatology.

Recent scientific studies have strongly suggested that pulsed biofilm and persister drug regimens, such as dapsone combination therapy, may be an effective solution for the management of CLD/PTLDS, along with the treatment of polymicrobial infections and multiple inflammatory triggers [14,30,38,39,71,72,73]. Emerging research indicates that biofilms are likely part of many, and probably most, chronic infections, providing a protected niche for pathogenic bacteria to survive, contributing to the burden of inflammation [74]. In our previously published research, dapsone therapy, using dapsone alone, or with combination antibiotic therapies, including a tetracycline, rifampin, and/or azithromycin, has demonstrated efficacy against the biofilm/persister forms of *Borrelia burgdorferi* [75]. Each additional intracellular antibiotic added to dapsone in culture increased the drug’s ability to reduce the mass, viability, and protective mucopolysaccharide layers of the *B. burgdorferi* biofilm [75]. The four-drug combination of dapsone, a tetracycline, rifampin, and macrolide (azithromycin) had the most significant effect on the *Borrelia* biofilm, and this was the protocol we used in our three case studies discussed here. Johns Hopkins researchers had also published that 6 days of combination antibiotic therapy using rifampin, azithromycin, and methylene blue were effective in killing *Bartonella* stationary-phase persisters, lowering the biofilm forms of *Bartonella henselae* in culture [76], completely eradicating the infection. Methylene blue was therefore added to our treatment protocol, both for its effect as a persister drug against *Bartonella* and *Borrelia* [76,77] and because of its effectiveness in lowering methemoglobin levels, a known side effect of dapsone, where oxidative stress impairs the oxygen-carrying capacity in the blood [78]. Finally, pyrazinamide, a tuberculosis drug used to decrease the length of treatment of mycobacterial infections by killing non-replicating persisters [79], was added when *Bartonella* was present, based on a prior published case study showing that the medication increased the efficacy of dapsone combination therapy in an autoimmune patient with Behçet’s disease [72].

Next-generation sequencing technologies have demonstrated that polymicrobial infections are also more frequently implicated in chronic infections now, often associated with biofilm formation, thus changing antimicrobial susceptibility patterns due to resistance and tolerance mechanisms [80]. Both *Borrelia burgdorferi*, the agent of Lyme disease, and *Bartonella* spp. have been discovered to be stationary-phase persister bacteria [81,82], located in biofilms [77,83,84], and co-infections of these two bacteria in ticks are not only being reported in the medical literature [85,86,87,88] but both bacteria have also been identified as co-localizing infections that may impact the severity of the infectious process [63,89,90]. This was the circumstance with the three patient case studies we outline below, who not only had symptoms of CLD but also active *Bartonella* infections, worsening their clinical status, along with evidence of exposure to other co-infections, including *Babesia* spp. All three patients also had exposure to COVID-19 during the course of their illness, along with evidence of exposure to environmental toxins, including heavy metals and mold toxins. Broad exposure to environmental toxins may impact susceptibility to these rising infectious diseases by increasing inflammation, weakening the immune system, and reducing vaccine effectiveness [91].

The three patients described in the following case presentations had a history of Lyme disease and *Bartonella* and/or *Babesia* with multiple abnormalities on the MSIDS map, and they finished at least one course of DDDCT. Patient 1 received one 6-day pulse of HDDCT after 8 weeks of DDDCT and remained in full remission for 12 months, after being ill for approximately 13 years. Patient 2, after following DDDCT, received a total of four 4-day HDDCT pulses, followed by two 6-day HDDCT pulses during a two-year time frame, which was completed in early January 2023, and despite being ill for over 4 years, they went from 15–20% of normal functioning to close to 100% normal functioning two months post treatment. Patient 3, who was ill for 15 years, after following an eight-week DDDCT protocol in December 2020, received five four-day pulses of HDDCT during the following two years. She gradually improved with each HDDCT pulse but did not go into full remission (without any further Lyme and tick-borne symptoms for 3 months or longer) until she received one 7-day course of HDDCT in 2023. She has now been in remission for over 12 months, despite contracting COVID-19 six times during a period of three years. A retrospective chart review of an additional 22 patients previously demonstrated that, in total, 100% of patients improved their tick-borne symptoms post DDDCT and HDDCT despite some having active co-infections (*Babesia*, *Bartonella*), especially if all overlapping MSIDS variables were adequately addressed [39].

These three female patients were part of a group of 25 patients who had previously completed the DDDCT and HDDCT protocols, outlined in our prior publication [39]. We assessed co-infection status, age, gender, length of illness, response to treatment, i.e., self-reported improvement in Lyme disease symptoms, and whether there was remission, percentage improvement, or lack of response to the HDDCT protocol. Full remission in our prior publication was defined as the resolution of all active tick-borne symptoms for at least three months post therapy. Partial remission was defined as the resolution of all active tick-borne symptoms for at least 6 weeks post therapy since some patients had recently finished the HDDCT protocol. All 25 patients in our retrospective chart review met the criteria for a clinical diagnosis of Lyme disease supported by a physician-documented erythema migrans (EM) rash and/or positive laboratory testing, including a positive ELISA/enzyme immunoassay (and/or C6 ELISA), immunofluorescent antibody (IFA), Centers for Disease Control and Prevention (CDC) positive IgM and/or IgG Western blot (WB), PCR, Borrelia-specific bands (23, 31, 34, 39, 83/93) on a WB [92,93], and/or positive ELISpot (lymphocyte transformation test (LTT)).

All patients signed informed consent forms that outlined the proposed benefits and potential risks of treatment; patients volunteered to take high-dose pulsed dapsone protocol at our medical center based on our prior research, illustrating the benefit of dapsone combination therapy in the treatment of CLD/PTLDS [14,38,71,73] and on the drug’s documented action on persister bacteria in biofilms [75]. None of the patients had a significant sulfa allergy or G-6-P-D deficiency, thus minimizing the possibility of allergic reactions or severe hemolytic anemia secondary to dapsone. Prior to beginning HDDCT, patients were required to have hemoglobin greater than 12 mg/dL, no active bleeding disorders, and no contraindication or significant allergies to any of the medications or supplements. The side effects of dapsone were explained in detail, including potential rashes, Herxheimer reactions, anemia, and methemoglobinemia [78,94]. The patients were asked to obtain a baseline methemoglobin level before starting DDDCT and pulsed HDDCT to ensure that there were no significant baseline elevations due to genetic variations or other medication interactions [95]. A complete blood count (CBC), comprehensive metabolic profile (CMP) with electrolytes, kidney and liver function, as well as methemoglobin levels, were obtained before, during (week 3, weekly on DDDCT and HDDCT), and post therapy (three to four weeks, as well as eight weeks after the completion of therapy) to ensure the reversal and normalization of any laboratory abnormalities. Healthcare staff closely monitored laboratory test results during and after DDDCT and HDDCT, and an emergency phone number was provided to all patients if urgent questions arose. A baseline EKG was also required prior to starting hydroxychloroquine, azithromycin, and/or ondansetron to rule out any associated QT prolongation [96] or arrhythmias, and patients were requested to repeat an EKG once all medications that could prolong the QT interval were on board.

Institutional Review Board approval was not required for this study since this was a retrospective review of a convenience sample of patient charts. A convenience sample of 25 adult patient charts out of a patient population of 50 patients who were offered HDDCT was chosen for inclusion in our study, and the three case studies outlined in this paper were among the 25 patients included in our prior publication [39]. We now provide a one-year follow-up in two out of three patients in order to demonstrate the long-term efficacy of DDDCT and HDDCT for CLD/PTLDS and associated infections, including *Bartonella*. A summary table, Table 1, which provides an overview of the most important laboratory and clinical variables that resulted in chronic illness and the number of dapsone combination therapies required to achieve remission, can be found below.

## 2. Case Presentations

### 2.1. Case 1

A 12-year-old white female, who was treated in our medical office with both parents’ consent, had a past medical history significant for Lyme disease, babesiosis, bartonellosis, orthostasis with frequent syncopal episodes, elevated liver functions with probable Gilberts, decreased ceruloplasmin and iron levels, human herpes virus 6 (HHV6), cytomegalovirus (CMV) and Coxsackie exposure, leukopenia with frequent episodes of serious otitis and pharyngitis with an adenoidectomy, and black mold exposure and presented to our medical office in December 2016 for an initial history and physical examination. Her chief complaints included sweats every 2 weeks, which were occasionally drenching, chills, moderate to severe fatigue, occasional hair loss, sore throat and swollen glands every three weeks with flareups, upset stomach with occasional constipation, rib soreness with rare chest pain, “air hunger” with some shortness of breath, rare palpitations, joint pain in the neck, back, and legs (calves and Achilles tendons) with associated neck cracking/stiffness, myalgias in the arms and legs, which would come and go, twitching of the face and eyes, rare headaches, tingling/numbness of the hands, stabbing pains in the shins and calves, occasional visual loss with blackouts and dizziness while standing/changing position, tremors of the right hand, moderate to severe memory concentration problems, severe brain fog, which would come and go, reversal of letters and numbers, insomnia with difficulty falling asleep and hypersomnia, and sleeping for 11 h without feeling refreshed. She became ill in 2010 after being in Missouri. Her illness began 5 weeks later and required evaluation by 51 medical doctors before she received a diagnosis of Lyme disease. At the time of her initial visit, she had been on cefdinir 300 mg PO BID, grapefruit seed extract, clarithromycin 500 mg PO BID, sulfamethoxazole/trimethoprim DS, and atovaquone/proguanil (Malarone) for two years without antibiotic rotations, reporting an improvement from 5% of normal functioning to 65% normal functioning after one month of treatment. She then plateaued without further improvement, with flareups of significant flulike symptoms, fatigue, pain, and brain fog every 3 weeks. She had tried amoxicillin initially, with minimal improvement, but did have an improvement in her *Babesia* symptoms, with a decrease in sweats while on atovaquone/proguanil. She needed to intermittently stop her medications secondary to low white cell counts (leukopenia), where she would relapse and be bedbound secondary to a flareup of her underlying symptomatology. At the time of her initial visit, she was on herbal therapy for Lyme disease, *Babesia*, and *Bartonella* (Byron White AL, A-Bart, and A-Bab) with *Cryptolepis sanguinolenta*.

Her social history was unremarkable, without any significant allergies, apart from a possible dermatological reaction to certain sunscreens. Her family history was positive for heart disease, thyroid cancer, and breast cancer, with one of her brothers suffering from asthma. A review of systems was essentially negative for significant cardiovascular, pulmonary, genitourinary, or gastrointestinal symptoms, although she had been hospitalized at 8 years old for significant elevations in liver functions and required a liver biopsy, without receiving a firm diagnosis as the cause of her elevated transaminases. She still complained about chills, occasional visual blackouts with dizziness while standing, syncope × 3, easy bruising, significant fatigue, joint pain in the neck, back, and lower extremities, occasional headaches, and significant memory/concentration problems. A physical examination revealed a well-developed and well-nourished 12-year-old white female in no apparent distress. She was 5′2″ tall, 87 pounds, and afebrile, with a respiratory rate of 16 per minute. Sitting blood pressure was 84/40, with a pulse of 92 and regular. Her standing blood pressure at 3 min was 84/52, with a pulse rate of 104 beats per minute (bpm); her standing blood pressure at 6 min was 84/52, with a pulse rate of 100 bpm. Her low blood pressure with an increase in heart rate of 12 bpm standing was consistent with the diagnosis of moderate postural orthostatic tachycardia syndrome (POTS). The rest of the physical examination was essentially within normal limits except for thin reddish crescents in the back of the throat, slight redness of the right tympanic membrane, mottling and purplish skin of the lower extremities, and several hemangiomas of the face and hands.

A record review revealed a normal CAT scan and MRI of the head in 2013, a normal spinal tap in 2013 with normal red cells, white cells, and glucose levels, negative oligoclonal bands, and slightly lower CSF protein (less than 10 (normal range: 12–60 mg/dL)), a normal MRI of the abdomen in 2016, an ultrasound of the abdomen revealing reactive lymph nodes in the right lower quadrant, a negative liver biopsy for elevated copper levels despite low ceruloplasmin, and a negative endoscopy in 2013. The medical records reflected black mold found in the master bedroom, without testing to evaluate the level and types of molds. Prior significant laboratory values included a positive Lyme immunofluorescent antibody (IFA) at 1:80, positive *Borrelia* specific bands on an IgM Western blot (23 kDa, 31 kDa, 34 kDa 93 kDa), positive *Borrelia* specific bands on an IgG Western blot (39 kDa), a low positive *Bartonella* IgM (1:20), negative *Babesia* titers and *Babesia* FISH, negative titers for *Ehrlichia*, *Anaplasma*, and *Rickettsia* (Rocky mountain spotted fever and Q fever), positive MTHFR (heterozygous), an elevated level of calcitriol (1,25 hydroxy vitamin D) at 92.1 (normal range: less than 75) with a low 25 hydroxy vitamin D (20.5, normal range: 30–96 nanograms/milliliter [ng/mL]), intermittent elevated calcium levels (10.5, normal range: less than 10.4 mg/dL) with a normal parathyroid (PTH) level at 36 pg/mL (normal range: 15–65), low iron saturation at 17% (normal range: 20–55%), low iron levels at 21 µg/dL (normal range: 26–169), low ferritin levels at 6 ng/mL (normal range: 15–77), mild leukopenia (white blood cell count 3.11, normal greater than 5.7/μL) with low segmented neutrophils at 0.72 (normal range: 2–5.8 × 10^3^/µL), elevated AST at 106 U/L (normal range: 10–40 U/L), and an elevated alkaline phosphatase level at 469 (normal range: between 36–128 U/L) with a mildly elevated total bilirubin (1.7, normal range: less than 1.2 mg/dL), consistent with genetically acquired Gilbert’s syndrome. CMV IgG antibodies were elevated, at 7.9 U/mL (normal range: less than 0.59 U/mL), and HHV 6 titers were elevated, at 8.3 (normal range: less than 0.76) with negative EBV titers.

Laboratory work requested included a complete blood count (CBC), comprehensive metabolic profile (CMP) with liver functions (LFTs), iron, ferritin, ceruloplasmin, alpha-1 antitrypsin level and genetic analysis, antibodies versus hepatitis A, B, and C, hemoglobin A1c, B12, folate, methylmalonic acid (MMA), homocysteine level (HC), antinuclear antibody (ANA), rheumatoid factor (RF), anti-double-stranded DNA antibodies, Smith antibodies, anti-scleroderma antibodies, Sjogren’s anti-SS-A, SS-B antibodies, ribonucleoprotein (RNP) antibodies, anti-Jo-1 antibodies, HLA DR 2/4 status, antiganglioside antibodies, anti-myelin antibodies, complement studies with C3, C4, C3a, and C4a, GAD 65 autoantibodies, an IgE food allergy panel with histamine, tryptase, and prostaglandin D2 levels, immunoglobulin levels and subclasses, red blood cell (RBC) and serum mineral levels (magnesium, copper, zinc, and iodine), serum glutathione levels, thyroid functions with antithyroid antibodies, insulin growth factor 1 (IGF-I), vaso-intestinal peptide (VIP), melanocyte-stimulating hormone (MSH), anti-diuretic hormone (ADH), serum pregnenolone and DHEA sulfate levels, and a repeat tick-borne panel to evaluate antibodies against *Borrelia*, *Babesia*, *Bartonella*, tularemia, typhus, and *Brucella*. A dehydroepiandrosterone sulfate (DHEA)/cortisol saliva test was given to the patient to perform at home to evaluate her adrenal function, along with a 6-h urine dimercaptosuccinic acid (DMSA) challenge and serum heavy metals (lead, mercury, and arsenic) to evaluate heavy metal exposure, as well as a RealTime Laboratory (Carrollton, TX, USA) mycotoxin assay to evaluate mold exposure.

Laboratory work returned positive for an elevated level of hemoglobin A1c at 5.7% (normal range: less than 5.6), consistent with metabolic syndrome/prediabetes and probable reactive hypoglycemia; low serum levels of alpha-1 antitrypsin at 87 mg/dL (normal range: 90–200); an alpha-1-antitrypsin (AAT) mutation analysis, revealing one copy of the Z allele (MZ), consistent with alpha-1 antitrypsin deficiency as a cause of the significant elevations in liver functions with transaminitis in the past, which led to a biopsy; elevated levels of bilirubin at 1.4 (normal less than 1.2), consistent with Gilbert’s syndrome; elevated levels of alkaline phosphatase at 353 IU/L (normal range: 134–349), consistent with a growth spurt; a low immunoglobulin G level at 610 mg/dL (normal range: 759–1549) with two subclass deficiencies (IgG subclass 1, 329 mg/dL [normal range: 456–952]; IgG subclass 2, 137 milligrams/dL [normal range: 147–493]), and low Immunoglobulin IgM at 49 mg/dL (normal range: 57–209), consistent with chronic variable immune deficiency (CVID) secondary to Lyme disease. HLA DR 2/4 was positive, increasing the risk of severe Herxheimer reactions and treatment of refractory Lyme arthritis [97]. There was also an elevated level of prostaglandin D2 at 367 pg/milliliter (normal range: 35–115), consistent with possible mast cell activation syndrome (MCAS); low plasma copper level at 0.63 µg/milliliter (normal range: 0.81–1.75) with a low ceruloplasmin level at 14.7 mg/deciliter (normal range: 19–39); normal serum copper level (75 µg/deciliter, normal range: 72–166), ruling out Wilson’s disease as a cause of elevated LFTs; low total iron (Fe) at 24 mcg/dl (normal range: 27–164); high iron binding capacity at 497 (normal range: 271–448 mcg/dL) with a low Fe saturation at 5% (normal range: 15–45%), consistent with Fe deficiency, ruling out hemochromatosis as a cause of elevated LFTs; low DHEA sulfate at 50.9 micrograms/deciliter (normal range: 67.8–328.6) with low normal cortisol levels in the morning and at night (a.m. cortisol 10, optimal range 14–25; evening cortisol 0.83, optimal range 1–4) through Labrix/Doctor’s Data (St. Charles, IL, USA), consistent with phase 2 adrenal dysfunction; and elevated GAD 65 antibodies at 97.3 U/milliliter (normal range: 0–5) with low C4 levels at 8 milligrams/deciliter (normal range:h 14–44). *Borrelia hermsii* IgG antibodies were positive at 1:64 (normal range: <1:64, Bioreference laboratories), consistent with possible prior exposure to relapsing fever *Borrelia*, and *Bartonella henselae* IgM antibody titers were low positive at 1:20, with a negative PCR (polymerase chain reaction). Mycotoxin levels came back significantly elevated through RealTime Laboratories (Carrollton, TX, USA), with an ochratoxin level of 46.516 ppb (parts per billion) (normal range: less than 1.8 ppb). Aflatoxins were elevated, at 5.969 ppb (normal range: less than 0.8 ppb). Trichothecenes were elevated, at 0.22 ppb (normal range: less than 0.02 ppb), and gliotoxins were elevated, at 5.229 ppb (normal range: less than 0.5 ppb). A 6-h urine DMSA challenge for heavy metals through Doctor’s Data revealed an elevated lead level at 8.3 µg/g creatinine (reference interval less than 3).

We discussed with the patient and family that Lyme disease, clinical babesiosis, probable bartonellosis, postural orthostatic tachycardia syndrome (POTS)/dysautonomia, phase 2 adrenal dysfunction with metabolic syndrome/reactive hypoglycemia, alpha-1 antitrypsin deficiency, mold, and heavy metal toxicity, along with chronic variable immune deficiency (CVID), were the primary diagnoses based on her record review, laboratory work, and physical examination that accounted for her chronic persistent symptoms. She was instructed to receive a baseline pneumococcal titer and check pre-/post vaccination titers for the immunodeficiency, and we rotated her antibiotic protocol to doxycycline, rifampin, Bactrim DS, and Malarone since she had never used a tetracycline and rifampin regimen for Lyme and *Bartonella* during the past 2 years. Regarding the babesiosis, she had also never used clindamycin or herbs including artemisinin and NEEM. She was therefore started on Plaquenil (hydroxychloroquine) 200 mg PO once a day (QD) (lower levels were used for her low body weight, and a recent eye exam was within normal limits), with doxycycline 100 mg PO BID, rifampin 150 mg (two capsules in the morning and one in the evening based on body weight), Bactrim regular strength one PO BID, nystatin 500,000 units 2 PO twice a day (BID), and three Malarone (atovaquone/proguanil) capsules per day with a high-fat meal, along with artemisinin SOD one PO TID. In an effort to control the history of severe Herxheimer reactions and inflammation, nutraceuticals that block NFKappaB were prescribed (N-acetylcysteine [NAC] 600 mg twice a day, alpha lipoic acid 600 mg once a day, and liposomal glutathione (Essential Pro, Wellness Pharmacy (Essential Nutraceuticals, LLC, Birmingham, AL, USA)) 500 mg once a day) along with supplements that stimulate the Nrf2 pathway, including turmeric (Curcuplex CR, Xymogen, Orlando, FL, USA) 500 mg twice a day and sulforaphane glucosinolate 100 mg once a day (Oncoplex ES, Xymogen). She was instructed to alkalize her body with sodium bicarbonate (NaHCO_3_) 2 g PRN for a severe Herxheimer reaction, along with 2000 mg of glutathione all at once. A low carbohydrate/hypoglycemic diet was prescribed to reduce the risk of yeast infections and due to her history of metabolic syndrome with mid-day fatigue with decreased stamina and concentration, along with triple probiotics (Theralac, Master Supplements (Victoria, MN, USA), Ortho Biotic (Ortho Molecular Products (Stevens Point, WI, USA)), and *Saccharomyces boulardii* (Ortho Molecular Products) twice a day. One gram of salt 3× per day with licorice extract was prescribed for low blood pressure and POTS/dysautonomia, along with drinking at least eight glasses of water per day and a mild physical therapy/conditioning program. The patient was instructed to buy an electronic home blood pressure monitor and follow her sitting and standing blood pressure and pulse rates daily; after sitting for 5 to 10 min after checking her sitting blood pressure and pulse rate, she was instructed to then stand, and her parents checked the readings at 3 min, 6 min, and 9 min standing and sent us the reports after 2 weeks, with any associated symptomatology. For insomnia, she was given a GABA-L-theanine cream.

The patient and her mother had a consultation one month later. Despite rotating the antibiotic protocol, night sweats and chills persisted, consistent with ongoing babesiosis, with severe fatigue. Doxycycline, rifampin, and Bactrim were discontinued (DC’d) and she was started on clindamycin 300 mg PO BID, increasing the dose as tolerated, working up to 900 mg per day, while remaining on azithromycin 250 mg PO BID, pulsed four days in a row per week. Malarone was also continued at a dose of 2 PO BID with a high-fat meal, and artemisinin was added with one capsule PO BID, with cryptolepis 30 drops twice a day. Three days after starting the protocol, she developed diarrhea, and all antibiotics were stopped, with bowel function returning to normal within 24 h. Probiotic dosing was increased to 3× per day, including *Saccharomyces boulardii*, Theralac, and VSL3. Coartem (lumefantrine/artemether) four tablets twice a day × three days was instituted for the ongoing symptoms of babesiosis as she had previously failed atovaquone (Mepron), along with one-half tablet of fludrocortisone (Florinef) 0.1 mg/day for ongoing low blood pressure as her blood pressure continued to remain hypotensive with dizziness while standing, despite drinking eight 8-ounce glasses of water, with 1 g salt tablets TID and Licorice Plus (glycyrrhizic acid, Metagenics). Cefuroxime axetil (Ceftin) 500 mg PO BID with sulfamethoxazole/trimethoprim (Bactrim) and nystatin 500,000 units, two tablets PO BID were eventually added after one week, but one month later, there was no significant change in symptoms. Post Coartem, she suffered from ongoing drenching sweats every 10 days and significant Herxheimer reactions resulting from the change in protocol, with increases in fatigue, joint pain, neuropathy (increased tingling and numbness), and brain fog. The patient was instructed in the future to use 2000 mg of liposomal glutathione for her Herxheimer reactions, and since she had never tried either dapsone or rifabutin, the Ceftin and Bactrim were DC’d, and she was rotated to doxycycline 100 mg PO BID, rifabutin 150 mg PO BID, Plaquenil (hydroxychloroquine) 200 mg QD, and dapsone 25 mg QOD × one week, and then, the dose was increased to 25 mg per day × one week. Rifabutin was eventually changed to rifampin 150 mg PO BID secondary to cost. As the dapsone was tolerated, she initially increased the dose of dapsone to 25 mg one day and 50 mg the next day, working up to a final dose of 50 mg per day, but ended up pulling back the dose of the dapsone to 25 mg every other day one month later due to significant Herxheimer reactions, which decreased her flares and led to an increase in energy.

The potential side effects of dapsone were discussed in detail with the patient and her family before instituting therapy, including anemia, Herxheimer reactions, rashes (if sulfa-sensitive), and methemoglobinemia (low oxygen-carrying capacity). Leucovorin (folinic acid) 25 mg QD, along with 15 mg of l-methylfolate (Folafy ER, Xymogen), was added to help counteract the dapsone-induced anemia, and N-acetylcysteine 600 mg twice a day, 600 mg of alpha lipoic acid, and 500 mg of liposomal glutathione were continued to support a healthy inflammatory pathway and help lower methemoglobin levels. Biofilm agents included Stevia (NutraMedix, Jupiter, FL, USA) 15 drops twice a day, Serrapeptase one capsule twice a day (NutraMedix), and Biocidin (Bio-Botanical Research, Watsonville, CA, USA), working up to two sprays twice a day. She was sent to an immunologist at National Jewish to evaluate the need for IV immunoglobulin therapy (IVIG) and was instructed to remain on a strict hypoglycemic diet for her metabolic syndrome and mid-day fatigue. Black mold was ultimately found in the master bedroom and remediated, and we explained that gliotoxins were immunosuppressive, overlapping her immunodeficiency with CVID. An oral mold protocol was therefore begun with 3 g (a teaspoon) of phosphatidylcholine (Phosphaline 4:1, Xymogen) twice a day, continuing her intake of NAC, alpha lipoic acid, and glutathione and adding low-dose WelChol (colesevelam) 625 mg, two tablets a day with the afternoon meal, at least four hours away from her oral medication to avoid decreased absorption. A multivitamin (Mitocore, Orthomolecular, Stevens Point, WI, USA) with fat-soluble vitamins was added to replace any potential deficiency secondary to the WelChol.

One month later, the patient reported significant improvements on low-dose dapsone (25 mg QOD) with doxycycline 100 mg PO BID, rifampin 300 mg PO BID, and Malarone two PO BID, including an increase in energy, decreased joint and muscle pain, neuropathy, and twitching, and increased cognitive functioning with decreased reversal of letters and numbers. *Babesia* symptoms with sweats decreased to one time per week, with rare chills, although she complained about pain in the soles of her feet with migratory back pain, consistent with ongoing symptoms of Lyme disease and *Bartonella*. She began intravenous immunoglobulins (IVIG) for her chronic variable immune deficiency (CVID) and was functioning at 85% of the normal level until she began the immunoglobulin therapy, which resulted in Herxheimer reactions, increased headaches, and a slight decrease in functioning. Methemoglobin levels remained low, at 1.6, and there was no significant anemia on dapsone. Postural orthostatic tachycardia syndrome (POTS), however, was not well-controlled despite salt, fluids, and Florinef as the patient’s sitting blood pressure of 96/54 with a pulse rate of 87 bpm went to 95/54 with a pulse rate of 136 bpm at 9 min standing. The 40-point increase in pulse rate with dizziness while standing and presyncopal episodes indicated severe POTS; so, fludrocortisone was decreased to half a pill of 0.1 mg per day, and midodrine was initiated at a dose of 2.5 mg three × per day (Q 4 h), along with metoprolol XL 25 mg QD for her tachycardia. She was instructed to continue daily sitting and standing blood pressure and pulse rates, and if her systolic blood pressure remained below 110 mmHg with ongoing postural orthostatic changes, she was instructed to increase the midodrine to 5 mg three × per day and start metoprolol XL (Toprol) 25 mg QD if tachycardic episodes continued. Adaptanall (Ortho Molecular) at two capsules once a day in the morning was begun for her adrenal fatigue, along with a low-dose glandular Adrenal Complex (PHP products).

Two months later, during an in-person history and physical examination, the patient continued to demonstrate significant improvement, having increased her dapsone dose to 25 mg per day along with doxycycline 100 mg PO BID, Plaquenil 200 mg QD, rifampin 300 mg BID, nystatin 500,000 units 2 PO BID, leucovorin 25 mg QD, l-methylfolate 15 mg QD, and Malarone 100/250 mg 2 PO BID. Her biggest improvements were in energy and cognitive function, receiving an A grade in all her school subjects, and fewer joint pains and flulike symptoms. Joint pain did persist in the neck, back, and legs, which would come and go, but the patient would now go for days without any pain for the first time in years. There was less dizziness while standing, fewer pre-syncopal episodes or visual blackouts on fludrocortisone (Florinef) ½ mg per day and midodrine 2.5 mg TID, and no palpitations on metoprolol XL 25 mg QD. There were also reduced tremors in the right hand, with no further tingling/numbness or stabbing pains (neuropathy) in the shins or calves and no twitching and headaches. Although the visual loss and blackouts from her postural orthostatic tachycardia syndrome (POTS) diminished, purplish mottling of the extremities persisted, consistent with dysautonomia. Her standing blood pressure at 10 min was 88/62 (mild orthostasis) with a pulse of 72 and regular. *Babesia* symptoms also persisted as she still complained about mild night sweats and moderate body aches with some swelling of her face post IVIG. Midodrine was therefore increased to 5 mg TID for her ongoing POTS symptoms while she continued to monitor sitting and standing blood pressure and pulse rates at home once a day and continued to detoxify her mycotoxins. Since more mold was found in the bathroom of her home, the glutathione was increased to 500 mg PO BID, and phosphatidylcholine was continued at a dose of 3 g twice a day. Her mother felt that the mold remediation and detoxification ultimately made a big difference in her symptoms, along with the low-dose dapsone combination therapy, and short sessions of far infrared sauna therapy (15 min) would be performed to assist detoxification if her POTS symptoms remained stable.

Three months later, in November 2017, the patient reported ongoing improvements in joint pain, no neuropathy, improved brain fog, and no significant POTS symptoms as blackouts had completely stopped. Her fludrocortisone was therefore DC’d, and midodrine was increased from 5 mg TID to 7.5 mg PO TID. Her energy level was better and stable but was her worst symptom, especially, with heavy physical exertion (running) or if she ate too many carbohydrates or missed meals, secondary to severe blood sugar swings with reactive hypoglycemia. Since she suffered from phase II adrenal fatigue, which can impact stamina and hypoglycemia, her dose of adrenal adaptogens was increased using Adrenal Complex (PHP products), one in the morning and one at 2 PM. Plaquenil 200 mg PO QD, doxycycline 100 mg PO BID, rifampin 300 mg PO BID, and dapsone 25 mg QD, with leucovorin 25 mg once a day and biofilm agents continued to be her best protocol to date, except for ongoing sleep issues, which would interfere with fatigue. She also felt that the IVIG was helping her to feel better, along with her mold detoxification. As her symptoms continued to improve, with no anemia (hemoglobin: 12.1 and hematocrit: 37.5) and a normal CMP with normal kidney and liver functions, the dapsone dose was increased to 50 mg one day and 25 mg the next day (an average of 37.5 mg per day), and if tolerated without significant Herxheimer reactions, it would then be increased to 50 mg per day while increasing leucovorin to 25 mg PO BID with 5 milligrams of l-methylfolate (MTHF-ES, Xymogen) to help limit dapsone induced anemia.

In February 2018, the patient noticed an improvement in symptoms on 50 mg of dapsone × 2 months, but in the interim, since November 2017, she had a repeat syncopal episode with low blood pressure, requiring the re-addition of fludrocortisone 0.1 mg per day, while increasing her midodrine to 10 mg PO TID and adding Licorice Plus (Metagenics) with glycyrrhizin for extra blood pressure and adrenal support. Her subsequent blood pressure levels improved, without further syncopal episodes. She also felt that mold detoxification with colesevelam, phosphatidylcholine, and glutathione improved her energy. She continued this protocol until June 2018, with continued improvement in her energy level, joint and muscle pain, and cognitive functioning, although still not near 100% normal functioning. Since her hemoglobin was stable at 10.5 g/dL and she continued to tolerate the dapsone combination therapy, we gradually increased the dapsone to 75 mg per day for one week and eventually increased it to 100 mg per day. In June 2018, a repeat hemoglobin was 10.0 g/dL (normal range: 11.9–15.5) with normal iron and ferritin levels, but a methemoglobin returned borderline elevated levels, at 2.2% (normal range: up to 1.8%); so, folic acid dosing was increased by adding 15 mg of l-methyl folate twice a day (Folafy ER, Xymogen) to her leucovorin 25 mg PO BID, while methylene blue was started at a dose of 50 mg PO BID, increasing glutathione to 500 mg PO BID.

In August 2018, she felt 100% well for the first time in 10 years with no fatigue, joint pain, night sweats, or brain fog. She finished 100 mg of dapsone per day for one month, with Plaquenil, doxycycline, and rifampin, and since a repeat CBC showed a mild drop in hemoglobin to 9.3 g/dL with a low haptoglobin of <14 mg/dL (normal range: 30–200), dapsone was stopped along with her other antibiotics while continuing on leucovorin, l-methylfolate, probiotics, and biofilm agents. Her mother felt that dapsone combination therapy was her best protocol in years and that midodrine was also a “game changer” along with adrenal support. She had come back with a low morning serum cortisol, at 4.1 µg/dL (normal range: 6.2–19.4), normal DHEA sulfate, at 116.7 µg/dL (normal range: 67.8–328.6), and normal pregnenolone, at 56 ng/dL (<151), requiring an increase in adrenal support. By December 2018, after 5 months off all antibiotics, she remained at 95% of normal functioning, and her chief complaint was random “hot flashes” without sweats or chills and rare hip pain without any significant fatigue. Hemoglobin increased to 11.5 with a hematocrit of 35.3, which was within normal limits (WNL) with a normal CMP. There were no clear menstrual flares with a relapse of underlying Lyme disease and tick-borne symptoms, and she remained in remission for 7 months until February 2019. She still required midodrine 10 mg PO TID to stabilize her POTS and found that she necessitated the administration of her IVIG on time (feeling symptoms starting to emerge 3 days before administration) with adrenal support while needing to stay on a strict hypoglycemic diet to stabilize her energy level. During that visit, she was asked to resend a mycotoxin urine test.

In November 2019, although the patient was generally stable during the past year, she began experiencing a few random, migratory aches in her legs and feet, which emerged with exercise, along with increased fatigue and a return of brain fog around her menses. The midodrine for POTS and IVIG continued to help her energy level but was losing its effectiveness. She reported an overall percentage of functioning at 89%, with no further day or night sweats, but had a return of insomnia, requiring 2–3 h to fall asleep, which started at the beginning of the school year with an increased academic load. Laboratory values performed to evaluate overlapping etiologies driving symptomatology revealed a mild iron deficiency anemia with a hemoglobin level of 10.9 g/dL (normal range: 11.1–15.9), low normal hematocrit level of 34.6% (normal range: 34–46.6), MCV 74 femtoliters (fL, normal range: 79–97), low iron level of 24 µg/dL (normal range: 26–169), and high total iron binding capacity (TIBC), at 483 (normal range: 250–450 mcg/dL), with a low ferritin level of 6 ng/milliliters (normal range: 15–77), consistent with the onset of iron deficiency anemia associated with her menses. Ceruloplasmin levels also remained low, at 18.7 mg/dL (normal range: 19–39), with a borderline low plasma copper level of 0.76 µg/milliliters (normal range: 0.8–1.75), and a normal RBC copper level; so, the patient was subsequently placed on an over-the-counter iron supplement (325 mg) with vitamin C and encouraged to receive more iron and copper in her diet with occasional meat, leafy green vegetables, seeds, nuts, and whole grains. Her CMP was normal, except for a slightly elevated alkaline phosphatase level consistent with a growth spurt and a low glucose level of 64, consistent with ongoing hypoglycemia. Dietary recommendations and compliance with a strict hypoglycemic diet were again discussed. Thyroid functions remained WNL, Epstein-Barr virus (EBV) and human herpesvirus 6 (HHV-6) titers were positive, consistent with prior exposure to EBV and HHV-6, with an EBV antibody VCA IgG at 72.6 units/milliliters (positive ≥ 21.9), EBV nuclear antigen antibody IgG at 76.1 unit/milliliters (positive ≥ 21.9), positive HHV6 IgG antibodies at 18.26 (positive ≥ 0.99), and negative EBV PCR, negative HHV-6 PCR. Repeat vitamin D levels revealed a low 25 hydroxy vitamin D level at 20.7 ng/milliliters (normal range: 30–100) but an elevated calcitriol level (1,25 dihydroxy vitamin D) at 67.9 pg/milliliters (normal range: 19.9–79.3), consistent with possible ongoing active intracellular infections. All autoimmune markers, i.e., anti-nuclear antibodies (ANA), double-stranded DNA (dsDNA), and rheumatoid factors (RF), remained negative, with normal inflammatory markers, i.e., erythrocyte sedimentation rate (ESR) and C-reactive protein (CRP), but she continued to have low complement C4a serum levels at 8 mg/dL (normal range: 14–44), consistent with immune activation from chronic Lyme disease and/or mold, low normal B12 at 350 picograms/mL (normal range: 232–1245), requiring B12 supplementation (Methylprotect, Xymogen), and normal histamine and chromogranin A levels but elevated prostaglandin D2 levels at 367 pg/milliliters (normal range: 35–115), consistent with possible mast cell activation syndrome (MCAS) from Lyme disease and mold toxicity. The patient did not notice any significant symptoms of elevated histamine (i.e., itching, sneezing, congestion, wheezing, etc.). Alpha-1 antitrypsin levels continued to be low, at 92 mg/dL (normal range: 100–188), consistent with alpha-1 antitrypsin deficiency, and heavy metals in the blood returned WNL (lead, mercury) with a borderline arsenic level of 6 µg/L (normal range: 0–23).

At a follow-up in-person visit in January 2020, she continued to notice an increase in Lyme disease symptoms with menstrual flares, constant fatigue, mild joint aches in the elbows, shoulders, hips, and knees, with stiffness of her neck and back, dizziness while standing, brain fog on and off throughout the day, slightly blurry vision, cold hands, and continued difficulty falling asleep. Symptoms continued to flare up around her menstrual cycle. She evaluated her level of normal functioning as slightly decreased from 89% to 83%. Her resting blood pressure was 117/65 with a pulse of 81 regular, which, at 5 min standing, was 104/71 with a pulse of 107 bpm, and at 10 min standing, was 112/70 with a pulse rate of 111 bpm. The increase of 30 bpm in pulse rate was consistent with ongoing severe POTS while remaining on midodrine 10 mg TID. Florinef 0.1 mg per day was re-added to her protocol with salt tablets 1 g, 3× per day, increasing her fluid intake from 4 to 5 glasses per day to 8 to 10 glasses per day (she was living at a high altitude). A 2-g dose of IV glutathione cleared her vision within several minutes, consistent with ongoing neurotoxins. We reordered a DHEA/cortisol adrenal test and mycotoxin test for the ongoing fatigue, pain, blurry vision, and brain fog, along with repeat CBCs, CMPs, thyroid functions, and Lyme disease and co-infection testing. The adrenal test continued to show phase 2 adrenal fatigue with a low normal morning cortisol level of 8.7 nanomoles per liter (optimal range: 14–25 nmol/L), low normal noon cortisol level of 3.1 nmol per liter (optimal range 5–10), a low evening cortisol level of 0.99 nmol per liter (optimal range 2–5), and normal DHEA level. The mycotoxin test revealed increased levels of ochratoxins, at 46.516 ppb (normal range: <1.8 ppb), increased levels of aflatoxins, at 5.969 parts per billion (normal range: <0.8 ppb), elevated levels of trichothecenes, at 0.22 ppb (normal range: <0.02 ppb), and elevated gliotoxins, at 5.229 ppb (normal range: <0.5 ppb). The trichothecene levels had decreased significantly (4.42 ppb → 0.22 ppb), as had levels of gliotoxins (20.32 ppb → 5.229 ppb). Lyme disease testing revealed an increase in *Borrelia*-specific bands. Her Lyme Immunoblot IgM had a new 23 (OspC) kDa band and decreased 31 kDa (Osp A), and her Lyme Immunoblot IgG showed a slight decrease in the 41 kDa banding (2+ to one plus). Lyme Multiplex genomic and plasmid PCRs were negative, and a *Bartonella henselae* immunofluorescent antibody (IFA) IgM and IgG were negative with a negative vascular endothelial growth factor (VEGF), at 33 pg/mL (normal range: 0–115).

Based on the above symptomatology and positive testing, we discussed going back on an antibiotic regimen, since she had been off all antibiotics for the past 18 months, and asked her to consider minocycline and rifampin and to use Bactrim instead of dapsone. The patient did not feel that she was sick enough to go back on antibiotics; so, we followed her and added Florinef 0.1 mg back to her protocol for the POTS and increased her mold detoxification. Her new protocol used phosphatidylcholine 3 g twice a day, glutathione 500 mg twice a day, N-acetylcysteine (NAC) 600 mg twice a day, and alpha lipoic acid (Alamax, Xymogen) 600 mg twice a day, and we rotated her WelChol to a different binder with G.I. Detox once a day (BioBotanical Research), containing activated charcoal/zeolite clay/silica/apple pectin and humic powder. The patient was instructed to take it two hours away from all medication and supplements to avoid binding any medications or other supplements. The family was also encouraged to recheck their home for any ongoing mold, and adrenal support was increased, using a glandular Adrenaliv (Xymogen) one at lunch with adaptogenic adrenal herbs (Adrenal essence, Xymogen) 2× per day.

During the follow-up consultation in July 2020, the patient had improved during the past 4 months, with less myalgias and brain fog (but still present), mild fatigue, and moderate sleep issues. She had a sudden episode of severe gastrointestinal pain with nausea the month prior, with increased RBCs in the urine, and was diagnosed with a new kidney stone by ultrasound, with a borderline elevated calcium level of 10.5 mg/dL (normal range: 8.9–10.4) and normal parathyroid (PTH) level. The analysis of the stone revealed calcium oxalate; so, the patient was placed on a low oxalate diet with increased fluids. Her OB/GYN also placed her on a birth control pill to regulate her menstrual cycle, but since she was having some increased acne and mild weight gain, we stopped the fludrocortisone. Immunoglobulin levels on IVIG were within normal limits, and all labs remained stable, except for new *Parvovirus B19* exposure (IgG level 6.4, positive ≥ 1.1) and a borderline positive *Mycoplasma pneumonia* IgG antibody at 104 U/mL (negative < 100). As of November 2020, the patient continued to do very well without antibiotics, with increased energy levels, less joint pain (knee pain was persisting), and increased sleep, but was weary of COVID, feeling stressed out and “grey” from the pandemic, and asked to see a therapist, which was beneficial.

In March 2021, the patient, who was now 17 years old, was taken off her birth control pill because of ongoing weight gain, bloating, and some fluid retention and was given a low-dose diuretic, i.e., hydrochlorothiazide (12.5 mg PRN), which helped fluid retention. Leptin levels and thyroid functions returned within normal limits, and sex hormone levels were normal (estrone, estradiol, estriol, testosterone, and DHEA), except for a low progesterone/estradiol ratio. She complained about ongoing joint pain, myalgias, neck stiffness, slight lightheadedness while standing, and mild cognitive difficulties, with memory/concentration problems, mood swings, and disturbed sleep, although she rated her functioning at approximately 91% of normal. She was now 2 ½ years without any further antibiotics for her tick-borne illness post lower-dose dapsone combination therapy. Based on her mild chronic symptomatology, we sent a repeat Lyme Immunoblot IgM and IgG, Lyme Multiplex PCR, and *Bartonella* FISH to IgeneX laboratories (Malpitas, CA, USA). Her Lyme Immunoblot IgM returned CDC-positive (23, 41 kDa), her Lyme Immunoblot IgG revealed a new 23 kDa (Osp C) and a 66 kDa band, her Lyme PCR was negative, and her *Bartonella* FISH returned positive, despite prior *Bartonella henselae* antibodies being low positive, with negative PCR and VEGF testing.

I discussed the results with the patient and her mother. Despite the positive Lyme Immunoblot and *Bartonella* FISH returning positive, she was still performing well in school and was not ready to restart antibiotics. Her biggest complaints were mild cognitive difficulties with memory and concentration problems, verbal dyslexia, mild muscle/joint pains, and, in particular, a weight gain of 17 pounds, despite being on a low-dose diuretic. We discussed undergoing a five-hour glucose tolerance test with insulin levels, along with an ultrasound of the ovaries to rule out polycystic ovarian syndrome (PCOS) if the weight gain should persist despite a strict hypoglycemic diet and being off her birth control pills and fludrocortisone. Her POTS was still severe, with pulse rates increasing by 37 beats per minute standing; so, she was encouraged to stay on the midodrine, lower her salt intake secondary to the water weight gain, and begin vagal/limbic retraining with either the Annie Hopper dynamic neural retraining system (DNRS) [98] or Gupta mindfulness-based retraining system [99]. Laboratory results showed an improved CBC (hemoglobin: 12.1 g/dL and hematocrit: 38%), with an MCV of 77 (microcytic), still suggestive of mild iron deficiency, normal free T3, normal free T4, normal TSH, normal sex hormone levels, low normal cortisol in the morning, at 7.1 µg/dL (normal range: 6.2–19.4 µg/dL), normal sedimentation rate and CRP, and normal fasting insulin level of 15.3 μIU/mL (normal range: 2.6–24.9). The leptin level was elevated, at 22, however, with a BMI (Body Mass Index) of 21.5 (upper range of normal 17.2 based on Tanner Stage), indicating possible leptin resistance.

By May 2021, the weight gain had not abated, with associated pitting edema (1–2+), and she began developing acne. We discussed starting minocycline 100 mg PO BID for her tick-borne disease and acne, along with rifampin 300 mg PO BID, Plaquenil 200 mg PO BID, nystatin 500,000 units 2 PO BID with triple biofilm agents, and triple probiotics. Fluvoxamine (Luvox) would have been considered for leptin resistance if the weight had not changed significantly. Unfortunately, the patient was unable to tolerate the antibiotics secondary to gastrointestinal upset and had to stop them after 6 days. By July 2021, the excess water weight had resolved with the regular use of a diuretic, and she was instructed to take hydrochlorothiazide with low-dose potassium every other day until she could see an endocrinologist for a second opinion. In the meantime, her OB/GYN switched her to a different birth control pill, Junel 1/20 (norethindrone/ethinyl estradiol), which helped decrease her heavy menstrual cycles and acne. POTS symptoms were also in remission with no dizziness while standing and no presyncopal or syncopal episodes. In November 2021, her primary symptoms were fatigue and weight gain, and we retried a low-dose doxycycline regimen (50 mg PO BID) for acne and low-grade tick-borne symptoms. We discussed following a double dapsone combination therapy protocol with a week of pulsed high-dose dapsone during the summer when she was off from school to treat her chronic Lyme disease and active *Bartonella*.

In the meantime, she developed COVID symptoms in December 2021, just as nirmatrelvir tablets and ritonavir tablets (Paxlovid) were being made available under the Emergency Use Authorization (EUA) [100], and she took several nutraceuticals to help lower inflammation by blocking NFKappaB, stimulating Nrf2, and blocking NLRP3 inflammasomes. These included increasing her doses of glutathione (2000 mg twice a day), N-acetylcysteine (NAC) 1200 mg PO BID, alpha lipoic acid 600 mg PO BID, curcumin (Curcuplex CR, Xymogen) 500 mg PO BID, sulforaphane glucosinolate (Oncoplex ES, Xymogen) 100 mg PO BID, zinc 20 mg PO BID, vitamin C 1000 mg PO BID, 500 mg of 3,6 beta glucan (Immunotix, Xymogen) with melatonin 1 mg at bedtime, and K2 D3 (5000 IUs, Xymogen) [101,102]. She denied having any active COVID symptoms despite the positive rapid test.

In April 2022, several months post COVID, tick-borne symptoms increased slightly, with a return of night sweats several times per month, increased motion sickness, moderate fatigue, and joint pain in the arms, legs, and hips several times per week (which would flare up during her menstrual cycle), along with word finding issues and moderate difficulty thinking. Her weight was stable, and midodrine 10 mg TID was controlling her POTS symptoms. We discussed different options, including starting with a *Babesia* protocol for the return of night sweats with tafenoquine 100 mg, twice per day for 3 days and then once a week for the next 6 weeks, followed by double-dose dapsone combination therapy and one week of high-dose dapsone combination therapy for her active *Bartonella* if symptoms persisted. The side effects of the protocols were discussed, and she and her mother signed an informed consent. A baseline CBC, CMP with mineral levels, VEGF, alpha-1 antitrypsin, and ceruloplasmin levels with immunoglobulins were checked prior to treatment. Her CBC had significantly increased to 13.9/40.3 by November 2022, secondary to iron replacement, with a normal CMP, except for a mildly elevated bilirubin at 1.2, consistent with Gilbert’s syndrome; ceruloplasmin increased to 41 mg/dL (normal range: 22–50), and her immunoglobulins and subclasses were now within normal limits on IVIG.

Double-dose dapsone combination therapy (DDD CT) was started in January 2023, lasting 2 months (Plaquenil, minocycline, rifampin, dapsone, and nystatin), with the addition of azithromycin 250 mg PO BID and pyrazinamide 1500 mg QD with methylene blue for her *Bartonella*, slowly increasing doses to 300 mg PO BID over several weeks. Each antibiotic was added slowly and taken with a full meal, after breakfast and dinner, for GI tolerance. During the last week of her protocol, i.e., week 9, she was instructed to take a 6–7-day pulse of high-dose dapsone combination therapy (HDDCT), i.e., 200 mg PO BID, while increasing her rifampin to 600 mg PO BID. She was to add nitrofurantoin 100 mg PO BID if active *Bartonella* symptoms persisted and to increase glutathione to 2000 mg all at once with Alka Seltzer Gold (NaHCO_3_) prn for Herxheimer reactions with high-dose leucovorin 25 mg, three capsules twice a day, with Folafy ER (15 mg l-methylfolate), three capsules twice a day, to reduce dapsone-induced anemia. This regimen included four biofilm agents (Biocidin, Stevia, cinnamon/clove/oregano oil (Doctor Inspired Formulations, Hopkinton Drug, Hopkinton, MA, USA), and compounded peppermint oil (Infuserve America, St. Petersburg, FL, USA)), along with four different probiotics (Ortho Biotic, Theralac, *Saccharomyces boulardii*, and Probiomax 350 billion, Xymogen, Orlando, FL, USA). A CBC, a CMP, and methemoglobin levels were checked in week 3 and then weekly on double-dose and high-dose dapsone combination therapy.

The patient finished the 9-week protocol early in March 2023, with 6 days of HDDCT. Her CBC in early February 2023, mid-protocol, was normal with a hemoglobin of 13.1, hematocrit of 39.9, normal CMP and liver functions (LFTs), and a normal methemoglobin level of <1.0%. At the end of February, her hemoglobin had dropped to 11.0, with a hematocrit of 33.5 (2.1 g drop in hemoglobin), with normal LFTs, except for a mildly elevated total bilirubin at 1.5 (normal range: 0.2–1.1 milligrams/dL) and a normal methemoglobin level. In early March 2023, after finishing the protocol, her hemoglobin was 10.9 and her hematocrit was 32.8 within an MCV of 99.1 (mild macrocytic anemia secondary to dapsone), with a normal methemoglobin level of 2.6% (normal range: 0–2.9%). Three weeks post protocol, after staying on leucovorin and l-methylfolate, her hemoglobin increased to 12.7, her hematocrit was 37.7, and her methemoglobin was <1.0, with mildly elevated LFTs (AST 36, normal range: to 32 U/L; AST 52, normal range: up to 32 U/L). She was instructed to increase her detoxification support and use Liver Protect (Xymogen) with milk thistle, NAC, glutathione, and alpha lipoic acid. As of May 2023, her laboratory values returned to normal with a hemoglobin of 13.2, hematocrit of 40.1, and MCV of 84.1, with a normal CMP and normal liver functions.

As of December 2023, the patient, who was now 19 years old, reported feeling “wonderful”, the best she had felt in the past 11 years, at 100% normal functioning. She reported significant increases in energy levels, improved cognitive functioning with no further dyslexia, no further joint pain, or neuropathy (tingling was gone), decreased hair loss, and stable weight. She denied any severe Herxheimer reactions on this protocol, and as of January 2024, the patient was approximately one year in full remission without any active Lyme disease or *Bartonella* symptoms and was able to attend her first year of college.

### 2.2. Case 2

A 21-year-old white female presented to our medical office for a Lyme disease consultation in May 2022. Her past medical history was significant for a history of Lyme disease, *Babesia*, and *Bartonella*, a movement disorder, with tremors and possible “pseudo seizures” (an electroencephalogram (EEG) ruled out grand mal, partial seizures, and/or a temporal seizure disorder), gastroparesis with intermittent nausea and vomiting, postural orthostatic tachycardia syndrome (POTS) and associated complaints of lightheadedness while standing, palpitations, fatigue, cognitive difficulties, anxiety, B12 and Vitamin D deficiencies, supraventricular tachycardia (SVT) with heart rates up to 180 beats per minute (bpm) (having been ruled out for Wolf-Parkinson-White (WPW) and Long-Ganong-Levine (LGL) syndrome as an explanation for the SVT), mold exposure without mycotoxin testing, possible reactive hypoglycemia with mid-day energy level crashes, and anxiety, paranoia, and OCD on sertraline (Zoloft) 50 mg per day, along with aripiprazole (Abilify) 10 mg per day, which were inadequate to control her symptoms, an abnormal endoscopy with *Candida* (white patches were noted in the gastric region), along with possible mast cell disorder, impacted wisdom teeth, acne, and plantar fasciitis. The patient had complaints of pain in the bottom of her right foot, and an MRI revealed fluid in that area, consistent with atypical plantar fasciitis, possibly related to her Lyme disease and *Bartonella*.

Chief complaints during her initial presentation included occasional night sweats, moderate fatigue, moderate joint pain in the knees and hips that were migratory in nature, intermittent myalgias in the lower extremities, mild headaches, tingling and stabbing sensations of the lower extremities bilaterally, which would come and go with foot pain, intermittent tinnitus, lightheadedness while standing, moderate cognitive difficulties with word-finding problems, occasional stuttering, severe anxiety, obsessive–compulsive disorder (OCD), rare palpitations, episodes of belching with hiccups interspersed with episodes of yawning, occasional difficulty moving her extremities, thinning hair, and a history of a movement disorder with possible migraines and auras based on prior neurological testing.

The patient had been in good health until age 17, approximately four years prior, when she reported a probable tick bite in France while playing soccer. She did not see a rash, and it took approximately 10 months to be diagnosed with Lyme disease and associated co-infections when she returned to the US for further diagnostic evaluations. Physicians in France, including a neurologist, had diagnosed her with cluster headaches and prescribed various medications, including propranolol (Inderal) and triptans (sumatriptan), which were ineffective. Another neurologist in Texas diagnosed her with migraine headaches and increased the Inderal dose and changed the triptan. None of these were helpful, and she was sleeping up to 20 h per day when she initially became ill with severe headaches, extreme fatigue, swollen glands, and painful, migratory arthritis. Her first Lyme disease protocol by a US physician included doxycycline and atovaquone/proguanil (Malarone), but she had significant nausea with the doxycycline and had to change the medication. There were also significant Herxheimer reactions with doxycycline, as well as cefuroxime axetil, which resulted in severe confusion, fatigue, and increased arthralgias, and over time, her condition worsened. By November 2018, she was admitted to a hospital in Texas for a week for possible seizures and myoclonic movements. She was seen by the chief of Infectious Diseases and had an MRI, a CT scan, and a spinal tap, which confirmed prior evidence of Lyme disease. The physicians, at that point, however, did not recommend any further treatment because she had taken oral antibiotics for over one month and because they had diagnosed her with a functional neurological disorder. She promptly returned to Paris and was admitted to the American Hospital of Paris for loss of balance, difficulty walking, and myoclonic seizure-like activity. An EEG was performed, which ruled out a classic seizure disorder. In July 2019, she returned to Texas as she was not improving and was given intramuscular (IM) ceftriaxone (Rocephin) twice weekly, with minocycline 100 mg twice a day. This improved her cognition and other symptoms, although she found that IV ceftriaxone was more effective than IM ceftriaxone. She had been accepted to a US university at that point but was too ill to continue her college education.

She subsequently saw a Lyme knowledgeable provider in Texas, who treated her for Lyme disease and *Bartonella*. She took different cocktails of antibiotics during the next several years, including hydroxychloroquine (Plaquenil), rifampin, azithromycin, pyrazinamide, and methylene blue for two positive T Lab *Bartonella* FISH tests. Due to gastrointestinal (GI) upset and a lack of adequate efficacity, she was rotated over to IV medications and received nine months of IV rifampin, IV azithromycin, and pulse IV ceftriaxone, along with eight months of disulfiram. She had previously tried dapsone in March 2020, with 100 mg for eight weeks, but had difficulty increasing the dose to 200 mg secondary to episodes of supraventricular tachycardia. Prior to this, she had been diagnosed with POTS and was told to increase her fluids and use scopolamine patches. Unfortunately, none of the above regimens were highly effective, and she developed progressive psychiatric symptoms; a psychiatrist put her on aripiprazole (Abilify) 10 mg for severe anxiety, psychosis (she was seeing monsters), and paranoia. This was helpful as her symptoms improved. One of the last pharmaceutical regimens, which were tried before coming to our medical office, was IV gentamicin, along with 16 months of oral and IV treatment for Lyme disease and *Bartonella*, but despite these protocols, she still tested positive for active *Bartonella* by a FISH test through T Lab, although there had been some clinical improvements. At the time of her presentation to our medical office, her symptoms were relapsing off antibiotics for the past three months, including moderate fatigue, moderate joint pain in the hips and knees, which were migratory, neuropathy of the lower extremities with foot pain, consistent with plantar fasciitis, moderate cognitive difficulties, and anxiety.

Her social history was unremarkable (no caffeine, cigarettes, or alcohol), and her medication included aripiprazole (Abilify) 10 mg QD, sertraline (Zoloft) 50 mg QD, and birth control pills, i.e., levonorgestrel 0.15 mg/ethinyl estradiol 30 mcg (Kurvelo). Medication intolerance/allergies included chlorhexidine (hives and swelling) and cefuroxime axetil (Ceftin), which caused a severe Herxheimer reaction. Her family history was unremarkable, except for a father who suffered from myelofibrosis, and both siblings (two other sisters) were alive and well. There were some sibling stressors since she was unable to continue her schooling while her siblings continued to advance educationally and professionally.

A review of systems was significant for occasional random night sweats, which significantly improved since treatment for her *Babesia*. Rare symptoms of allergic rhinitis, occasional palpitations, and dizziness while standing without presyncopal or syncopal events seemed to be related to the history of anxiety and POTS and two episodes of supraventricular tachycardia in the past, having been ruled out for conduction abnormalities. Gastrointestinal symptoms included increased bloating with carbohydrates, possible candidiasis, and a history of gastroparesis with nausea and vomiting (but no recent episodes for several months). Rheumatological symptoms included migratory pain in her hips and knees (consistent with active Lyme disease), as well as myalgias, primarily in the quadriceps region, which were moderate in intensity, with pain in the bottom of her right foot, consistent with plantar fasciitis. Neurological symptoms included mild headaches (which had improved with treatment), moderate cognitive difficulties, with memory/concentration problems, neuropathic symptoms of the lower extremities, a movement disorder, dizziness while standing, and anxiety. The sertraline and aripiprazole had been helpful in controlling psychosis and anxiety.

A physical examination revealed a well-developed and well-nourished white female in no apparent distress. She was afebrile, with a height of 5 feet and 5 inches and a weight of 150.6 pounds. Her respirations were 16 and regular. Her blood pressure was 149/84 sitting, with a pulse of 94 bpm and regular. Her standing blood pressure was 140/94 with a pulse of 100 beats per minute at three minutes: 142/96 with a pulse of 101 BPM at six minutes standing and 138/93 with a pulse rate of 93 beats per minute at 10 min standing. The decrease in systolic pressure of 11 points standing was consistent with mild POTS. Otherwise, a physical examination of the head, eyes, ears, neck, and throat, as well as cardiac, pulmonary, gastrointestinal, and neurological examinations were all negative. The only significant finding was dermatological: her skin had striae on her thighs, which were perpendicular to the skin planes, consistent with prior *Bartonella* exposure. There was no dermatographism.

Laboratory examination: abnormal laboratory results found in prior medical records included low B12 levels, at 254 pg/mL (normal range: 232–1245), low vitamin D levels, at 21.4 ng/mL (normal range: 30–100), prior Epstein-Barr virus exposure (EBV) with positive EBV antibody VCA IgG titers, at 431 U/mL (normal range: <18), positive EBV nuclear antigen antibodies > 600 U/mL (normal range: <18), negative EBV PCRs, a CDC-positive IgG Western blot (five positive bands: 18, 23, 41, 66, and 93 kda), two positive *Bartonella* FISH testing through T Labs, and a negative *Babesia* FISH test. Tick-borne testing conducted through our medical office ruled out exposure to *Ehrlichia*, *Anaplasma*, *Rickettsia rickettsii*, *Coxiella burnetti*, *F. tularensis* or *Brucella*, and VEGF levels were WNL. Other negative testing included autoimmune markers (ANA, RF, anticardiolipin Ab, anti-myelin Ab, GAD-65 autoantibodies, and CPK), inflammatory markers (CRP, ESR, and TGF Beta-1), serum hormone levels (HbA1c, thyroids, adrenal, posterior pituitary hormones, i.e., VIP, and MSH), serum and RBC mineral levels, and celiac markers (anti-gliadin, tissue transglutaminase) with normal levels of immunoglobulins and subclasses. G6PD levels and natural killer (NK) cell/activated T cell levels were also WNL, with no evidence of active viral infections (HHV6). Borderline levels of aluminum were found in the serum (9 μg/L, normal range: 0–9), with negative mercury, lead, and arsenic levels, ruling out overlapping causes for her neuropathy and neuropsychiatric symptoms, along with low glutathione levels (159 μg/mL, normal range: 176–323 μg/mL), which, in part, may have explained by the severity of her prior Herxheimer reactions. Other abnormal laboratory results included a 95 IgG food allergy profile through LabCorp (Burlington, NC, USA), which revealed 61/95 positive reactions to foods, consistent with leaky gut, which may have contributed to her underlying inflammation.

We repeated tick-borne testing through the IgeneX laboratory, and the patient was given an adrenal DHEA/cortisol salivary test to rule out adrenal dysfunction, along with a mold test through RealTime Laboratories in Texas to rule out mycotoxin exposure. The IgeneX Immunoblot revealed evidence of prior exposure to Lyme disease (IgG Immunoblot: 23 kda+ (Osp C), with weak positive bands at 31 kda (OspA) and 93 kda); *Babesia* titers and *Babesia* FISH were negative (*B. microti*, *B. duncani*), and a *Bartonella* Immunoblot IgM/IgG showed exposure to *Bartonella* species at the genus level, with positive exposure to *B. quintana* and negative *Bartonella* FISH. Adrenal testing revealed severe adrenal insufficiency with stage 3 adrenal dysfunction, where all four cortisol levels were significantly below the normal range (morning cortisol level of 6.6 nmol/L, optimal range: 14–25; noon cortisol level of 0.97 nmol/L, optimal range: 2.1–14; evening cortisol level of 0.91 nmol/L, optimal range 1.5–8.0; and nighttime cortisol level of <0.33 nmol/L, optimal range 0.33–7.0). Mycotoxin testing was also positive for exposure to 3/5 mycotoxins: trichothecenes (0.1390, normal range: <0.07 ppb), gliotoxins (2.2890, normal range: <0.5 ppb), and zearalenone (0.9180, normal range: <0.5 ppb).

The assessment and plan, after reviewing the clinical presentation, physical examination, and laboratory testing was that the patient suffered from active Lyme disease due to her migratory joint pain and CDC-positive IgG Western blot with a spinal tap confirming prior evidence of Lyme disease, active *Bartonella* due to two positive *Bartonella* FISH tests through TLab with evidence of *Bartonella* striae, foot pain/plantar fasciitis, and ongoing neurological symptoms, including a movement disorder with severe neuropsychiatric symptoms of anxiety, OCD, and psychosis, despite almost 4 years of oral, IM, and IV antibiotics, along with B12 and Vitamin D deficiencies, severe adrenal insufficiency, mycotoxin exposure, mild to moderate POTS/dysautonomia with vagal dysfunction and gastroparesis, intermittent supraventricular tachycardia, possible reactive hypoglycemia/*Candida* and leaky gut with multiple food sensitivities, and possible mast cell disorder. For her B12 deficiency, she was given one methyl protect per day (Xymogen) with methyl B6, methyl B12, and methyl folate. She was given K Force (K2D3, 5000 IU, Ortho Molecular) for vitamin D deficiency, along with a strict hypoglycemic yeast-free diet, avoiding sensitive foods in a rotation diet, using bovine immunoglobulins and spore-based probiotics for her leaky gut. This consisted of Mega IgG 2000, two capsules twice a day (Microbiome Labs), and MegaSpore, one capsule twice a day (Microbiome Labs). She was also given adrenal support with adaptogenic herbs (Adrenal Essence, two capsules with breakfast, Xymogen) and an adrenal glandular capsule at breakfast and lunch (Adrenaliv, Xymogen) for her phase III adrenal dysfunction, slowly increasing the dose to two capsules twice a day, misoprostol (Cytotec) 200 mcg 3× per day with ondansetron (Zofran) 4–8 mg Q8 prn for her nausea and gastroparesis, and finally, valerian root (SynovX calm, Xymogen) 2× per day with GABA-L-theanine (CopaCalm, Ortho Molecular), 1–2 dropperfuls prn for her anxiety. Calm abiding meditation and biofeedback training (https://www.heartmath.com/, accessed on 21 April 2024) were also suggested for her anxiety and vagal dysfunction.

We discussed with the patient that multiple intracellular antibiotics using biofilm agents and persister drug regimens are usually necessary in order to effectively address chronic active Lyme disease and *Bartonella* (i.e., minocycline or doxycycline, azithromycin, rifampin, PZA, dapsone, and methylene blue) and that although she had taken disulfiram and dapsone in the past, based on published medical research, the doses would have been inadequate to control and effectively eliminate the infections. Also, due to her prior gastrointestinal problems with oral antibiotics, we would need to provide extra GI support prior to starting the protocol. She was therefore prescribed famotidine 10 mg, up to 20 mg twice a day, Cytotec (misoprostol) 200 µg twice a day, and GlutAloeMine (Xymogen) with glutamine, aloe vera, and deglycyrrhizinated licorice, one scoop twice a day, to support her gastrointestinal function and improve tolerability. We therefore started one antibiotic at a time to ensure GI tolerance and try and limit the possibility of severe neuropsychiatric Herxheimer reactions, along with low-dose naltrexone (LDN) 4.5 mg QD in the morning for neuroinflammation/microglial activation.

Her first regimen consisted of Plaquenil (hydroxychloroquine) 200 mg PO BID, minocycline 100 mg PO BID, and nystatin tablets 500,000 units to PO BID, taken together after breakfast and dinner. She was instructed to take minocycline on a full stomach, avoiding dairy products and minerals (magnesium, zinc, copper, iodine, etc.) within one hour, taking it with a full 8-ounce glass of water sitting up to avoid any possible reflux esophagitis. This was taken with grapefruit seed extract, two capsules twice a day, along with Plaquenil, to address the cystic/cell wall-deficient forms of *Borrelia*. For biofilms, she was prescribed cinnamon/clove/oregano oil capsules, one twice a day, Biocidin, two sprays twice a day, working up the dose slowly based on tolerance, Stevia drops, 15 drops twice a day, and peppermint oil capsules, one twice a day. For microbiome support, she began taking four different probiotics, upon awakening and at bedtime, one hour away from antibiotics, i.e., Theralac (Master Supplements, Victoria, MN, USA), one twice a day, Ortho Biotic (Orthomolecular), one twice a day, *Saccharomyces boulardii* (Ortho Biotic), one twice a day, and ½ a scoop of Probiomax 350 billion (Xymogen), one twice a day. Once this regimen was tolerated, after one week of taking minocycline, Plaquenil, and nystatin, while keeping a diary of any symptom changes while starting this regimen, she then added oral rifampin 300 mg twice a day to the Plaquenil, minocycline, and nystatin. We explained that rifampin can change the drug levels of her medication (psychiatric drugs and birth control pills) and that she needed to be in contact with her psychiatrist regarding tapering down her medications so that she could eventually start methylene blue. The psychiatrist began a slow 4-week taper of sertraline and aripiprazole so that by month 2, she could begin dapsone combination therapy. In week 3, once the minocycline and rifampin were well tolerated, she added pyrazinamide, 500 mg tablets, three tablets once a day, and in week 4, as GI tolerance was good without severe Herxheimer reactions, azithromycin 250 mg PO BID, was added 4 days in a row per week (Monday through Thursday).

During the second month of therapy, the patient began dapsone combination therapy with 25 mg of dapsone once a day for several days, increasing to 50 mg per day for several days, then increasing to 75 mg per day for several days, and holding at 100 mg per day as long as there were no significant Herxheimer reactions. A rapid increase in dosing was possible as she had already been on dapsone 100 mg in the past and tolerated it. It was only higher dosing that resulted in severe Herxheimer reactions with episodes of supraventricular tachycardia, but she had not been diagnosed with glutathione deficiency at the time of her first dapsone protocol (which would theoretically increase Herxheimer reactions), nor had she used a beta blocker on a regular basis for her POTS. Nutritional supplementation, therefore, included blocking NFKappa B with NAC 600 mg BID (which would also support glutathione production), alpha lipoic acid (Alamax CR, Xymogen) 600 mg PO BID (which helps regenerate glutathione while lowering down free radical/oxidative stress), and glutathione 500 mg PO BID (Orthomolecular), gradually increasing the amount in the second month as the dapsone dose increased. Nrf2 activation, to help support a healthy inflammatory response, was begun with turmeric (Curcuplex CR, Xymogen), one twice a day, and sulforaphane glucosinolate 100 mg PO BID (Oncoplex ES, Xymogen); low-dose melatonin, ½ tablet of a 3 mg dose (Xymogen, melatonin CR) was used at bedtime to block NLRP3 inflammasomes, a third inflammatory pathway. These nutraceuticals were taken with folic acid support, i.e., leucovorin (folinic acid) 25 mg, two in the morning and one in the evening (total of 75 mg per day), along with L-methyl folate (Folafy, Xaquil XR, and Xymogen) 15 mg, two in the morning and one in the evening. The methylene blue was slowly begun in week 5, at a dose of 50 mg PO BID, once dapsone dosing was at 100 mg per day, and a low dose beta-blocker, i.e., metoprolol XL, 25 mg per day, was added secondary to the prior episodes of supraventricular tachycardia that previously resulted from higher-dose dapsone combination therapy. She was instructed to complete four weeks of double-dose dapsone combination therapy (100 mg of dapsone twice a day, i.e., DDDCT), followed by her first 4-day pulse of high-dose dapsone combination therapy (200 mg of dapsone twice a day, i.e., HDDCT) for *Bartonella* at the end of the therapy.

As of June 2022, one month into the treatment, although the patient noticed Herxheimer reactions with moderate fatigue, brain fog, body spasms, and continuation of her movement disorder, there were marked improvements in anxiety, headaches, and neuropathy, which was now non-existent, and her *Babesia* symptoms (night sweats) had resolved, along with an improvement in tinnitus. During the second month on double-dose dapsone combination therapy (DDDCT), however, the azithromycin had to be DC’d secondary to stomach upset, although she was able to continue with the minocycline, rifampin, PZA, and dapsone. Cimetidine 400 mg PO BID was therefore added in month two to help manage potential symptoms of methemoglobinemia and GI upset, along with famotidine 10 mg PO BID, antioxidants (Vitamin C, Vitamin E, glutathione, and NADH), and methylene blue. There were no symptoms of SVT during DDDCT, and she was encouraged to try re-adding azithromycin 250 mg PO BID in the last week of therapy with a 4-day pulse of HDDCT to treat her chronic *Bartonella*, along with hydroxychloroquine 200 mg PO BID, rifampin 600 mg PO BID, minocycline 100 mg PO BID, PZA 1500 mg QD, dapsone 200 mg PO BID, and methylene blue 300 mg PO BID.

She completed the first round of DDDCT and 4-day HDDCT pulse by August 2022, and nine days post completion, she went from 40% of normal functioning prior to therapy, to 70% of normal functioning. There had been improvements in fatigue and brain fog, which were now mild–moderate in intensity, with no headaches, rare neuropathy symptoms, less lightheadedness and anxiety, mild joint pain, and no night sweats, although she still complained about the loss of focus during conversations and mood swings, with moderate depression. Her seizure-like episodes/movement disorder had resolved for the first time in several years. She was advised to increase her AdrenaLiv further for phase III adrenal dysfunction if resistant fatigue persisted while staying on a strict hypoglycemic, candida-free diet and following a one-month mitochondrial regeneration with ATP 360, 3 QD (Researched Nutritionals), CoQ power 400 mg BID (Researched Nutritionals), acetyl-L-carnitine 2 PO BID (Carnitex, Xymogen), and NADH 5 mg QD (ENADA). During the period without antibiotics, in between future HDDCT pulses, a mold protocol was begun for her mycotoxin exposure, with phosphatidylcholine 2.5 g PO BID (BioPC Pro, Orthomolecular), glutathione 500 mg PO BID (Orthomolecular), glucomannan (Optifiber Lean, Xymogen), one scoop QD, 1 h away from medications and supplements, in between breakfast and lunch for trichothecene exposure, and G.I. Detox (BioBotanical Research) with charcoal and bentonite clay, 2 h away from medications and supplements, between lunch and dinner. All other supplements (NAC, Alamax, probiotics, biofilm agents, etc.) were continued.

In September 2022, one month post DDDCT and her first 4-day HDDCT pulse, the patient had been feeling 100% of normal functioning for the first time in years, and then several days prior to her menstrual cycle, the symptoms relapsed for 3 days, with a return of night sweats, fatigue, joint pain, and mood swings. These symptoms promptly resolved once she had her menses, where she became completely asymptomatic with no fatigue or mid-day energy level crashes and no myalgias, joint pain, headaches, neuropathy, lightheadedness, movement disorder, or brain fog. Her mood had also improved, but she decided to go back on sertraline 50 mg QD to stabilize her symptoms. By October 2022, two months post dapsone combination therapy, her laboratory values had almost returned to normal, staying on folinic acid (leucovorin) with L-methyl-folate in tapering doses. She had a hemoglobin level of 12.7 (prior Hb was 13.5 gm) and a hematocrit level of 38.4% (prior Hct was 40.7%), and her CMP was WNL. However, her symptoms began to relapse 2 months post treatment, with mild fatigue, mild joint pain (hips, as knees were improved), mild headaches, and mild-to-moderate cognitive issues, with difficulty falling asleep. Her ‘pseudo seizures’ also began to return with an intermittent movement disorder. We, therefore, discussed tapering off her sertraline and performing a second HDDCT pulse, slowly adding in Plaquenil 200 mg PO BID, minocycline 100 mg PO BID, nystatin 500,000 U PO BID, rifampin 300 mg PO BID, pyrazinamide 1500 mg QD, and azithromycin 250 mg PO BID, with gradually increasing doses of methylene blue in week one, working up to 200 mg PO BID. In the last week, dapsone was added at a dose of 100 mg PO BID × 3 days, for tolerance, finishing with a 4-day HDDCT pulse, i.e., dapsone 200 mg PO BID × 4 days, along with leucovorin 25 mg, 2 PO BID, L-methyl-folate 15 mg, 2–3 PO BID, with methylene blue 200 mg PO BID. Due to her phase III adrenal dysfunction, where her fatigue was especially prominent in the morning, we increased her adrenal glandular, Adrenaliv, to three in the morning and one in the evening to stabilize her energy level and improve the tolerability of the protocol.

In early December 2022, one month after her second 4-day HDDCT pulse, with MB at a dose of 200 mg PO BID, the patient was much better. Her level of functioning went from 30% to 95% of normal functioning. There was an improved energy level, with mild fatigue, no *Babesia* symptoms, no further headaches, arthritis, or neuropathic symptoms, and only mild brain fog, although there was an episode of vaginal yeast, secondary to some dietary indiscretions with sugar, which required an OTC antifungal (vaginal miconazole). When she had her menses at the end of the month, it felt like the movement disorder was going to return, but it never did, which was markedly improved since the month prior to treatment. During the one month off antibiotics, she decided to go back on aripiprazole 10 mg for her history of anxiety, although her prior psychiatric symptoms never returned, and in between her antibiotic pulses, she began a mold detoxification protocol for her mycotoxin exposure. She started phosphatidylcholine (BioPC Pro, Ortho Molecular) 2.5 g PO BID, NAC 600 mg PO BID, glutathione 500 mg PO BID, glucomannan (OptiFiber Lean, Xymogen), one scoop between breakfast and lunch, and G.I. Detox (BioBotanical Research) with bentonite clay and charcoal, two hours away from all medications and supplements. She also completed a mitochondrial regeneration protocol (ATP 360, 3/day; Co-Q Power, 1 BID; carnitex 2 BID; NADH 5 mg/day) with biofilm agents, probiotics, and adrenal support. She remarked that these helped her to feel better, especially the glutathione and adrenal supplementation, which stabilized her energy level. There was no further evidence of gastroparesis, and her POTS symptoms were better post therapy, although, if she rapidly changed position, she noticed some palpitations and dizziness. She was instructed to send us 10 days of daily sitting and standing BP and pulse readings and to drink at least eight 8-oz. glasses of water per day with extra salt, while remaining on a strict hypoglycemic diet.

In early January 2023, two months after completing the initial 8 weeks of DDDCT and her second 4-day pulse of HDDCT, she had been feeling great, but she traveled to South Africa and contracted a cold (not COVID), and her symptoms began returning after her viral exposure and menses. She experienced mild-to-moderate fatigue, mild migratory joint pain, occasional headaches, slight tinnitus, mild cognitive problems, and frequent episodes of yawning (30× the day prior, which was an old symptom), and her movement disorder began to return. There were still no night sweats, lightheadedness while standing (POTS symptoms were improved), or neuropsychiatric symptoms. Her adrenal support was increased to three Adrenaliv in the morning and two in the early afternoon, which improved her energy level, and her last counts from several months ago showed mild anemia with an Hb of 11.3 and Hct 36%. She was instructed to go back on her probiotics, taper the aripiprazole, and increase the iron in her diet while rechecking her labs and preparing for a third HDDCT pulse in mid-January 2023. In this round, her third 4-day pulse of HDDCT, she was instructed to increase her methylene blue dose from 200 mg PO BID to 300 mg PO BID for the first time.

In mid-February 2023, one month post therapy, the patient was feeling better, functioning at approximately 70% of the normal level. She tolerated the quadruple dapsone well, with only minimal Herxheimer reactions. There was mild improvement in fatigue, but she ran out of her adrenal supplements, which interfered with her progress. The joint pain was now only in her left hip, without migratory pain anywhere else, and there were small improvements in her cognition such that her brain fog was better, with no headaches, yawning episodes, neuropathy, or ‘pseudo-seizures’, although she described some unusual ‘tics’ with eye blinking twice a week during the month, which was a new symptom. There was no more stuttering, and her mood was good. However, she reported having another viral exposure right before receiving her third pulse, which had increased her symptoms. Her most resistant symptoms at this point in time were fatigue, which appeared to have a significant adrenal component, intermittent mild joint pain (3/10 intensity), and rare body ‘jerks’, although she was significantly better than 10 months prior, during her initial consultation. She felt she needed a fourth pulse for her resistant symptoms and working around her school schedule, and she decided to decrease the antibiotic pulse to an 11-day protocol, rather than 14 days, by quickly ramping up the minocycline, rifampin, PZA, and azithromycin in week 1 since she had not had any GI problems with the antibiotics in the past.

By March 2023, one month after her third 4-day pulse of HDDCT, with dapsone 200 mg PO BID and 300 mg of MB PO BID, the patient was “feeling really good”, up to 95% of normal functioning. She noticed that her fatigue was the only mild symptom that remained, although it continued to improve day by day and week by week. Her headaches and neuropathy were now gone, her sleep was better, there were no further yawning episodes or ‘pseudo seizures’, and her cognition was also 100% back to normal. Mild fatigue was the only remaining symptom, and to address the fatigue, she was to instructed stay on her adrenal support, folic acid, methyl B12, and Fe to reverse the dapsone-induced anemia and to complete a one-month mitochondrial regeneration protocol, taken after each pulse of dapsone combination therapy. Apart from using ATP 360, CoQ Power, carnitex, and ENADA, we now added enhanced mitochondrial support with MitoNR (Designs for Health), two a day, with Mitoprime (Xymogen), one a day, to help with the regeneration of glutathione.

The next follow-up was 6 months later, feeling what she labeled as ‘great.’ Although after the third HDDCT pulse, she went two months without symptoms, she now had a 4-month stretch after her fourth high-dose dapsone pulse, where she was asymptomatic, without Herxheimer flares around her menstrual cycle. However, two months ago, after being jet lagged while returning from South Africa, she began to “feel off”. Her mother had been sick during the prior 2 months with COVID-19 and long COVID symptoms, and it was unclear if the patient had silently contracted COVID-19 during this period of time. Residual *Bartonella* symptoms were returning, with increasing fatigue and a relapsing movement disorder. Abnormal myoclonic movements had now returned twice a week during the prior two weeks, although the intensity and frequency of the movements were less intense and less frequent compared to her initial clinical presentation. We discussed receiving another HDDCT 11–14-day pulse, but because she was back in school and feeling a lot better after the last round of treatment, she decided to hold off on the dapsone pulse until she finished her exams and had a break from school. She started back on probiotics to prepare her GI tract, and in early October 2023, she received her fifth dapsone pulse but her first 6-day HDDCT pulse.

During the November visit in 2023, 5–6 weeks after her high-dose 6-day pulse, she remarked that certain symptoms came out for the first time when she extended the HDDCT pulse from 4 to 6 days. During the pulse, neuropathy returned at bedtime, with intermittent numbness on the right side of her face, lasting several hours, and then going away. *Bartonella* striae also returned on her left shoulder and back, right before receiving her first 6-day high-dose dapsone pulse, confirming an active infection; however, the longer the timeframe from the last pulse, the better she was feeling. During her sixth week post treatment, her level of functioning went from 75% to 95% of the normal level. Her symptoms included mild fatigue and brain fog, mild, rare tinnitus, and a slight problem with her equilibrium, although her ‘pseudo seizures’/movement disorder had not returned during the past 6 weeks. There were also no further arthritic symptoms, and her mood was normal, with no further anxiety, OCD, or hallucinations. We told her to stay on a small dose of Abilify (aripiprazole) 2.5 mg/day while she was recovering and until she had her follow-up with her psychiatrist. By the end of December 20, 2023, approximately 2 ½ months post four 4-day HDDCT pulses and her first 6-day HDDCT pulse (five pulses in total), the patient was feeling 90% back to normal. Her arthritis, which usually returned during her menses, had stayed in remission, and she tapered off her Abilify with normal moods. Her only symptoms included slight residual fatigue (she was also traveling a lot), mild intermittent brain fog (which is presently gone), and mild intermittent neuropathy of the feet, which returned after her last menstrual cycle. She felt as if one more dapsone pulse would be sufficient to put her in remission and planned to start her second 6-day HDDCT pulse at the end of December 2023.

By the end of January 2024, it was approximately three weeks post dapsone for the patient. She now had completed DDDCT and four 4-day HDDCT pulses, with her second 6-day HDDCT pulse (6 days of 400 mg of dapsone with 600 mg of methylene blue). The patient reported that the “treatment was intense”, but the Herxheimer reactions were significantly less than the last round of treatment, with only two days of increased symptoms. The first week post dapsone, she was still recovering, but approximately three weeks post dapsone, the patient reported “I am feeling great! I have absolutely zero brain fog and overall I am feeling excellent. There were some moments (very few of them) when I had not felt well, but overall, considering it has only been three weeks, I can truly say that I have never felt better!” The patient waited several weeks to start the supplements post dapsone (high-dose folate and mitochondrial support) as she felt that she needed a break while traveling to NY but restarted them at the end of January, with a repeat laboratory evaluation in February 2024, with a CBC, a CMP, and methemoglobin level.

### 2.3. Case 3

A 35-year-old white female with a past medical history significant for chronic fatigue syndrome/fibromyalgia, phase 3 adrenal dysfunction, Raynaud’s phenomena, *Helicobacter pylori*, parasites (pinworms), acne/eczema, depression/anxiety with post-traumatic stress disorder (PTSD) secondary to a history of multiple episodes of sexual abuse, chronic gynecological issues including polycystic ovarian syndrome (PCOS), bacterial vaginosis, herpes simplex virus-1 + 2 (HSV-1, HSV-2), human papillomavirus (HPV), frequent urinary tract infections (UTIs) with pyelonephritis, requiring 18 months of nitrofurantoin (Macrodantin) PO QD, candidiasis, black mold exposure with mild elevations of mercury in the blood, low blood pressure with syncope, and reactive hypoglycemia with palpitations, presented to our medical office in May 2020 with a long list of medical complaints. Her chief complaints were severe, unrelenting fatigue, depression/anxiety, musculoskeletal pain with tension in the neck and shoulders, migratory neuropathy, insomnia, weight gain, severe memory/concentration problems, low-grade fevers with night sweats, shortness of breath, swollen glands and sore throats that would come and go, intermittent bladder symptoms with urinary tract infections (UTIs), intermittent stomach pain, occasional headaches/migraines, intermittent tremors, lightheadedness, palpitations, twitching of the eyelids, occasional double vision, lack of sleep, and tinnitus.

Her symptoms started in 2008 when she was living in Germany. She did not see a tick bite but was living in the woods while attending a small university. Her symptoms started with restless leg syndrome and aching muscles of the lower extremities, which subsequently spread to her arms in April 2008. This was followed by symptoms of tingling in the lower extremities, which then spread to the upper extremities and face. After studying abroad, she returned to the United States to study in a midwestern state, but physicians there were unable to provide a diagnosis or treatment to help her symptoms. One year later, she developed severe depression and anxiety after battling depression in high school secondary to episodes of bullying and sexual abuse. She did intensive therapy with a trauma counselor who did eye movement desensitization and reprocessing (EMDR) and was treated with duloxetine (Cymbalta), fluoxetine (Prozac), and escitalopram (Lexapro). In 2010, she then began to experience increasing fatigue, insomnia (problems falling asleep and frequent awakening), decreased memory and concentration, as well as chronic urinary tract infections, bacterial vaginosis, and candidiasis. She went to a major medical center in Texas, and secondary to her dizziness while standing with vertigo, she was checked for POTS, with a negative head-up tilt table test (HUT). She was suspected to have multiple sclerosis (MS) but had a negative MRI of the brain and spine. She was prescribed gabapentin for her neuropathy without a clear etiology and was told her symptoms were secondary to stress. She remained on gabapentin for two years, but it gradually lost efficacy over time, despite increasing the dose. She then moved to San Francisco, where she saw a Lyme disease knowledgeable physician, who diagnosed her with chronic Lyme disease, relapsing fever, *Babesia duncani*, and anaplasmosis. He also diagnosed her with *Helicobacter pylori* and was treated with nitazoxanide (Alinia) for six months after taking doxycycline for one week, which was stopped secondary to nausea and vomiting. The nitazoxanide caused weight gain and increased depression and anxiety; so, it was rotated to amoxicillin with clarithromycin and omeprazole for approximately one year, without any improvement. There were occasional Herxheimer reactions, with no improvement after stopping the medications. She then reached out to another Lyme knowledgeable physician in January 2020, who suggested disulfiram, but it was never started because of her mental instability and work stress. The patient was working as a venture capitalist but was disabled due to the severity of her symptoms, rating her level of functioning at 20% of the normal level.

During her initial history and physical examination, her social history was unremarkable. She was on no medication, except for diphenhydramine (Benadryl) PRN for sleep, with nutritional supplements, including NAC, glutathione, magnesium glycinate, cordyceps, melatonin, valerian tea, St. John’s wort, 5 hydroxy tryptophan (5 HTP), adrenal adaptogenic herbs, vinpocetine, and probiotics. Medication intolerances included vomiting with doxycycline, weight gain with nitazoxanide, and nausea with penicillin and sulfamethoxazole/trimethoprim (Bactrim). Amitriptyline (Elavil) was ineffective for sleep. Her family history was remarkable for melanoma (mother and father). Her work and social life were significantly impacted by her CLD and co-infections, leading to heightened psychiatric symptoms.

A review of systems was significant for low blood pressure with intermittent dizziness, occasional episodes of syncope, low blood sugar and reactive hypoglycemia with weakness after eating sugar, chronic candidiasis, low-grade fevers, chills, a 30-pound weight gain, intermittent sore throats with swollen lymph nodes in the axilla and groin, intermittent blurry vision/double vision with lack of sleep, tinnitus in the right ear, palpitations and shortness of breath, both at rest and with exertion, exercise-induced asthma (cold weather), frequent UTIs with one episode of pyelonephritis, intermittent outbreaks of HSV-1, HSV-2, HPV, and bacterial vaginosis, decreased libido, migratory myalgias and neuropathy, migratory joint pain in the wrists, fingers, hips, knees, and ankles, Raynaud’s phenomenon, eczema (elbows) with pruritus after certain foods (beer and wine), migraines/headaches around her menses, and striae which developed on her flanks, buttocks, and inner thighs. She had cats growing up, receiving multiple bites and scratches, but had never been diagnosed with *Bartonella*.

A physical examination revealed a well-nourished and well-developed white female in no apparent distress. The patient was 5 feet and 8 inches tall, afebrile, and 174.4 pounds, with a resting blood pressure of 132/87, a regular pulse of 98, and a respiratory rate of 18 per minute. There were no significant changes in blood pressure or pulse rate sitting or standing, consistent with POTS, and her examination was essentially within normal limits, with the exception of *Bartonella*-type striae on her flanks, buttocks, and inner thighs.

Abnormal results found in the medical records included a positive IgM Lyme Immunoblot 4/2018, positive *Babesia duncani* titer (1:40, increasing to 1:80 one year later), human granulocytic anaplasmosis (HGA) titers at 1:80, positive IgM *H. pylori* antibodies, positive *Candida* antibodies, mildly elevated serum mercury levels (6.7, normal range: <10), low vitamin D and B levels, mildly elevated eosinophils (CBC), elevated C4a, an abnormal CDSA with increased gram-negative bacteria, and phase 3 adrenal dysfunction with low cortisol levels in the morning, at noon, and in the evening. The testing ordered during the initial consultation included a CBC, a CMP, G6PD level, B12, folic acid, methylmalonic acid, homocysteine, methylenetetrahydrofolate reductase (MTHFR), iron, TIBC, ferritin, vitamin D level, ammonia level, natural killer (NK) cells (total and function), CRP, ESR, ANA, rheumatoid factor, creatine phosphokinase (CPK), HLA DR 4, antiphospholipid antibodies, antiganglioside antibodies, anti-myelin antibodies, mineral levels (magnesium, selenium, copper, and zinc), serum heavy metals (mercury, lead, arsenic, and aluminum), an IgE food panel, thyroid functions with anti-thyroid antibodies, insulin growth factor-1 (IGF-I), sex hormones (estradiol, progesterone, testosterone, DHT, free testosterone, and sex hormone-binding globulin), DHEA sulfate, pregnenolone, MSH, ADH, VIP levels, histamine, chromogranin A, tryptase levels, antigliadin antibody, TTG, total immunoglobulins and subclasses, matrix metalloproteinase-9 (MMP 9), C3a, C4a, *Stachybotrys* titer, glutathione levels, co-Q10 and carnitine levels, B vitamins, viral titers and PCRs (HHV-6, EBV, and West Nile antibodies), and a parasite panel with antibodies for *Entamoeba histolytica* (amebiasis), *Giardia*, *Strongyloides*, and schistosomiasis. Local tick-borne testing through Quest laboratories included a C6 ELISA, *Ehrlichia* and *Anaplasma* titers, *Bartonella henselae* titer, vascular endothelial growth factor (VEGF), Rocky Mountain spotted fever antibodies, Q fever, typhus, tularemia, and *Brucella* antibodies. IgeneX laboratory testing included a *Bartonella* Western blot, *Bartonella* FISH, and *Babesia* FISH.

Based on the patient’s history and laboratory work, she was diagnosed with chronic Lyme disease, relapsing fever *Borrelia*, active babesiosis with occasional night sweats, chills, low-grade fevers and shortness of breath, *Anaplasma*, probable active *Bartonella*, candidiasis, reactive hypoglycemia, possible occult POTS/dysautonomia due to her extensive neuropathy and dizziness while standing with syncopal episodes, depression, anxiety, premenstrual dysphoric disorder (PMDD), and PTSD secondary to trauma and abuse, chronic insomnia with restless leg syndrome, chronic UTIs with possible interstitial cystitis (IC) secondary to Lyme/*Bartonella*, intermittent outbreaks of HSV-1, HSV-2, and HPV, bacterial vaginosis, possible mold toxicity with mast cell activation and migraines, possible heavy metal toxicity with mercury and tinnitus, phase 3 adrenal dysfunction, allergic rhinitis with exercise-induced asthma, vitamin and mineral deficiencies, and possible mitochondrial dysfunction.

During the follow-up consultation one month later, laboratory testing returned positive for mineral deficiencies (iodine, zinc, and iron, with low ferritin), elevated heavy metals (aluminum), low glutathione levels, HLA DR 4 positivity with elevated autoimmune markers (ANA 1:80, anticardiolipin antibodies, antiganglioside antibodies, and phosphatidylserine antibodies), with an indeterminate *Bartonella* Western blot, and evidence of prior exposure to *Bartonella vinsonii* sub spp. The *Babesia* FISH and *Bartonella* FISH were negative, but tularemia and *Brucella* titers returned low level positive at 1:20, consistent with possible cross-reactivity with other intracellular infections (like *Bartonella*). We discussed starting with minocycline 50 mg PO BID after a full breakfast and dinner, working up the dose slowly secondary to her prior intolerance to doxycycline, with nystatin 500,000 units 2 PO BID and triple probiotics and then, after discussing the risks and benefits of dapsone combination therapy, we gradually added dapsone 25 mg PO QOD, increasing to 25 mg QD one week later if there were no significant Herxheimer reactions. Leucovorin was started at 25 mg PO QD with Folafy ER (15 mg of l-methyl folate) to address dapsone-induced anemia, and the patient was to add rifampin in the future for Lyme disease and *Bartonella* with methylene blue, biofilm agents, and potentially Macrodantin (Macrobid), depending on her GI tolerance. N-acetylcysteine 600 mg PO BID and glutathione 500 mg PO BID were prescribed for her glutathione deficiency, with a trial of 2000 mg of glutathione QD X 7 days to evaluate the role of neurotoxins with her blurry vision, along with a multivitamin with B vitamins and K2 D3 for vitamin deficiencies. Valerian root (Synovx calm, Xymogen), two capsules HS and 1–2 PRN, was used for sleep and anxiety, with or without L theanine 100 mg PO TID, along with cyclobenzaprine 5 mg HS PRN for sleep, and we discussed starting a regular meditation practice with her psychotherapy, along with a gentle exercise program. A low histamine diet was prescribed for her pruritus, with possible mast cell activation, which was beneficial, and a strict hypoglycemic diet improved blood sugar swings and helped stabilize energy levels. Lauricidin (monolaurin), one scoop/day, was added for biofilm/*Candida* support, we replaced her minerals (zinc, iodine, and iron) with zinc glycinate 20 mg PO BID, a multimineral supplement (MinRx and Xymogen), and FeSO_4_ 324 mg, one a day (taken away from antibiotics to avoid interfering with absorption), along with adaptogenic herbs for phase 3 adrenal dysfunction (Adapten-all, Ortho Molecular; Adrenaliv, Xymogen, two capsules upon awakening and two at 2 PM). A RealTime Laboratory mycotoxin test was given to the patient to do in the near future.

During the follow-up consultation in July 2020, the patient noted minor improvements in her symptoms on minocycline 50 mg PO BID, nystatin 500,000 units 2 PO BID, dapsone 25 mg QD, and leucovorin 25 mg QD, with fewer myalgias, swollen lymph nodes, nausea, vomiting, dizziness, and anxiety; however, fatigue, mood swings, insomnia, night sweats, and brain fog had worsened. She was also complaining about gastroesophageal reflux disease (GERD); so, famotidine 20 mg PO BID was prescribed. For her increased *Babesia* symptoms with sweats, dapsone was increased from 25 to 50 mg per day, with Malarone 100/250 mg, 2 PO BID, and Coartem for PO BID × 3 days while increasing her minocycline to 100 mg in the morning and 50 mg in the evening. Limbic retraining with Annie Hopper DNRS was helping her mood disorder, apart from regular psychotherapy, and for resistant insomnia, although cyclobenzaprine 5 mg HS was beneficial, it still took her 90 min to fall asleep with frequent awakening; so, Honokiol 2 HS with 200 mg of magnesium L-threonate (OptiMag Neuro, Xymogen) was added to her sleep regimen.

At the end of August 2020, her night sweats initially worsened and then improved, relapsing and remitting, with four pulses of Coartem and Malarone. Myalgias and neuralgias, lymph node swelling, insomnia, and brain fog with cognitive difficulties were mildly better with a higher dose of dapsone (50 mg) and the addition of Honokiol (now obtaining eight to nine hours of sleep per night), with less pain in the soles of her feet (*Bartonella*), but she still suffered from severe chronic fatigue, headaches, increased light and sound sensitivity, and tremors. Rifampin was therefore added to minocycline and dapsone at a dose of 300 mg, one PO BID, for resistant symptoms, and after one week on rifampin, she was to begin slowly increasing the dose of dapsone (75 mg per day for one week, increasing to 100 mg per day if tolerated) while concomitantly increasing her doses of folic acid with leucovorin 25 mg PO BID and l-methyl folate 15 mg PO QD. Rifampin was held while on pulse Coartem to avoid lowering levels of the medication, and *cryptolepis sanguinolenta* (Crypto plus, Researched Nutritionals), 1 teaspoon three x per day, with artemisinin SOD (Researched Nutritionals), was added for ongoing symptoms of babesiosis. During this time, her mood continued to improve, despite rifampin potentially lowering levels of her SSRI, which she attributed to DNRS, which was a “game changer”. A CBC, a CMP, and the methemoglobin level were checked when she was up to dapsone 100 mg per day, which showed no anemia or elevated levels of methemoglobin.

In November 2020, the patient had now been on dapsone 100 mg per day for 3 weeks with Plaquenil, nystatin, minocycline, rifampin, triple biofilm agents (Biocidin, Lauricidin, and oregano oil), and triple probiotics, and she noticed significant improvements in her underlying symptoms. There were fewer fevers, sweats, chills (*Babesia*), anxiety, depression, and gastrointestinal issues (no diarrhea or vomiting with GERD), improved energy levels and headaches, and fewer tremors, with better sleep, i.e., 7 to 8 h per night. She still complained about some mild pain in the soles of her feet and some carpal tunnel symptoms, working on a computer keyboard for five to six hours per day. An ergonomic keyboard was prescribed, which helped. Her worst symptoms included migratory joint pain in her wrists, hips, ankles, and fingers, along with neck and shoulder pain and jaw pain/muscle tension. A muscle relaxant, methocarbamol 750 mg, was prescribed, one PO TID PRN during the day, continuing the same doses of antibiotics. Extra adrenal support was added for 3 PM fatigue, increasing the dose of Adrenaliv to two capsules at 2 PM, along with a strict hypoglycemic diet. By November 2020, the patient’s level of functioning had increased from 25% of the normal level to 40% of the normal level, *Candida* was under control, and there were no further *Babesia* symptoms. Since the patient was stable, with normal hemoglobin (13.1 g/dL, normal range: 11.7–15.5), CMP, and methemoglobin levels (0.00), the plan was to receive her first course of DDDCT, increasing dapsone to 100 mg PO BID, with Plaquenil, nystatin, minocycline, and rifampin, while increasing her folic acid (leucovorin at 75–100 mg per day and Folafy ER at 30–45 mg) and adding methylene blue 50 mg PO BID, tapering off her selective serotonin reuptake inhibitor (SSRI) to avoid potential interactions (serotonin syndrome). Ondansetron (Zofran) 4–8 mg Q8 PRN was used if there was any nausea/vomiting, along with misoprostol (Cytotec) 200 µg PO BID if any gastric upset occurred.

In early December 2020, the patient finished her first course of DDDCT. Compared to 6 months prior, she had improved, with less fatigue, swollen lymph nodes, myalgias, light and sound sensitivity, GERD, and bladder pressure, improved cognition, and fewer mood swings (improved anxiety and depression), but she still complained about fevers, drenching night sweats HS, which were waking her up, secondary to resistant *Babesia*, restless leg syndrome (RLS), headaches, worsening joint pain in the wrists, hips, ankles, and fingers, pain in the soles of her feet (consistent with active *Bartonella*), tremors, and unbearable neuropathy, despite using gabapentin 300 mg PO TID. Her labs were stable post dapsone, with mild anemia, normal CMP, and no elevated levels of methemoglobin. Her hemoglobin decreased to 11.1 g/dL (normal range: 11.7–15.5)/hematocrit 33.7 (normal range: 35–45%); so, she remained on leucovorin 25 mg PO BID and Folafy ER once a day to reverse the anemia, with nystatin, probiotics, and biofilm agents. At this point, she had failed *Babesia* treatment with atovaquone, Zithromax, Coartem, Malarone, and antimalarial herbs but had never taken tafenoquine. She was prescribed tafenoquine 300 mg one time as malaria anti-relapse treatment and was told to contact our office in a week with the results. If the RLS symptoms were to continue, dopaminergic agents such as ropinirole would be considered.

In January 2021, one month post DDDCT, her laboratory values were within normal limits (normal CMP and methemoglobin), and her anemia had completely reversed on folic acid (hemoglobin of 14.7 g/dL and hematocrit of 44.2). She was improving weekly, with most symptoms having improved, except for weakness in her limbs, neuropathy, restless legs that would come and go, and pain in the soles of her feet. She had also failed tafenoquine as a single anti-malarial medication, with ongoing flushing and night sweats, and was not a candidate for clindamycin with her history of candidiasis; so, a repeat course of Coartem, three days on and 11 days off, with cryptolepis and antimalarial herbs, was prescribed, along with one month of a mitochondrial regeneration for her weakness (ATP 360, three per day, Researched Nutritionals; co-Q power 400 mg BID, Researched Nutritionals; acetyl-L-carnitine (Carnitex) 1000 mg PO BID, Xymogen; and ENADA (NADH) 5 mg one PO QD). Famciclovir (Famvir) 250 mg PO BID was prescribed for the suppression of viral infections as she was having intermittent herpes outbreaks, and she was instructed to continue obtaining sitting and standing blood pressure and pulse rates to rule out POTS with her ongoing weakness and history of dizziness with pre-syncopal events.

In February 2021, the patient developed COVID-19, which led to an increase in all her underlying tick-borne symptoms, including her sweats, chest tightness, fatigue, joint pain, brain fog, headaches, and neuropathy. She had improved until that point. The mitochondrial regeneration had helped her muscle weakness, and the nerve pain had been better controlled the month prior on a long-acting gabapentin, i.e., gabapentin enacarbil (Horizant) 600 mg PO BID. We discussed using a high-dose glutathione protocol (2000 mg PO TID) for 14 days for COVID-19, along with NAC 1200 mg PO BID, alpha lipoic acid (Alamax, Xymogen) 600 mg PO BID to block NFKappaB, Curcuplex (turmeric, Xymogen) 500 mg PO BID, and sulforaphane glucosinolate 100 mg PO BID (Oncoplex ES, Xymogen) to stimulate Nrf2, along with 1 mg of melatonin at bedtime to block NLRP3 inflammasomes, zinc 20 mg PO BID, vitamin C 1 to 2 g PO BID, K2 D3 (10,000 IU Vitamin D), and Immunotix (3, 6 beta glucan, Xymogen). Since her worst symptoms were consistent with active *Babesia* and *Bartonella*, we discussed a trial of a very low-dose Disulfiram 62.5 mg, every one to two weeks, avoiding all alcohol, staying at low doses to avoid significant Herxheimer reactions or increased neuropathy (immediately stopping for any neuropathic flares), with a maximum dose of 62.5 mg QD, considering adding back on low-dose dapsone for combination therapy. In order to keep down *Candida* and minimize antibiotic use, we pulsed Zithromax 250 mg PO BID 4 days in a row per week with methylene blue 50 mg PO BID and considered adding pulse rifabutin, 150 mg, 2 PO BID one day per week (for *Bartonella* persisters) if symptoms persisted, gradually working in minocycline and pyrazinamide if there was an inadequate response. Fluconazole would be used PRN on Sunday, several days off azithromycin if needed for any flareups of yeast.

In March 2021, the patient had improved to approximately 70% of normal functioning, but due to an increase in GERD with occasional vomiting, increased tinnitus, sound sensitivity (which was better the days off azithromycin), and bladder pressure, methylene blue and Zithromax were DC’d, and she was rotated to Plaquenil 200 mg PO BID, nystatin 500,000 units 2 PO BID, clarithromycin ER 500 mg PO BID with rifabutin 300 mg PO BID 2 days per week (Monday, Thursday). There had been no Herxheimer reactions on disulfiram 62.5 mg per day; so, the dose was gradually increased over the next 7 weeks to a target goal of 125 mg per day, immediately stopping if there were any significant increases in psychiatric symptoms or neuropathy. Biofilm support was extended with cinnamon/clove/oregano oil twice a day. Laboratory values remained within normal limits, with a normal CMP and normal liver functions, zinc and iodine levels, and hemoglobin/hematocrit levels off dapsone, continuing on leucovorin 25 mg PO BID. Dapsone 25 mg per day was therefore added back on for antimalarial support due to ongoing night sweats QHS. MinRX (Xymogen) with 400 mg of magnesium glycinate was added at bedtime for muscle spasms and jaw tension, and methylene blue was restarted at 50 mg PO BID as her bladder symptoms had improved. One month later, she improved to 80% of normal functioning with less flushing, myalgias, leg pain, breast pain, and joint pain in the hands and soles of her feet, along with decreased blurry vision, light and sound sensitivity, tremors, chest pain, nerve pain, and bladder pain. She still, however, suffered from drenching night sweats, severe fatigue, neck and shoulder pain, headaches, swollen axillary lymph nodes, and severe migratory joint pain in the knees, hips, ankles, wrists, and fingers, along with severe brain fog. Glutathione helped her cognition, but her processing speed was extremely slow. Due to the severity and recalcitrance of her *Babesia* symptoms, methylene blue and rifabutin were DC’d, and she was given higher-dose atovaquone 750 mg, 2 teaspoons PO BID with a high-fat meal with clindamycin 300 mg, 2 PO BID, ivermectin 13.5 mg QD (based on body weight), with artemisinin SOD (Researched Nutritionals) one PO TID, and compounded *cryptolepis sanguinolenta* 30 drops TID in peppermint oil (Infuserve America). She was told to restart azithromycin if symptoms persisted.

By May 2021, she was up to 85% of normal functioning, with many symptoms having improved, and she reported “feeling the best in a long time” but continued to suffer from drenching night sweats, resistant insomnia, and increased pain in the palms and soles of her feet, with a flareup of underlying symptoms 2 weeks post COVID vaccination. Clindamycin and dapsone were DC’d, along with folic acid supplementation, and azithromycin 250 mg PO BID was added to atovaquone 2 teaspoons PO BID with sulfamethoxazole/trimethoprim double strength (Bactrim DS) one PO BID with her prior *Babesia* herbs. Once her *Babesia* symptoms had improved, the plan was to re-add intracellular medications for *Bartonella* (minocycline, rifampin, and pyrazinamide). Unfortunately, her *Babesia* symptoms persisted, with drenching night sweats; so, Bactrim DS was DC’d, and dapsone was worked back up to 100 mg per day with disulfiram 62.5 mg per day, atovaquone, azithromycin, ivermectin, artemisinin, *cryptolepis sanguinolenta*, and biofilm agents. We discussed going for a yearly eye exam and seeing her OB/GYN to evaluate her hormones as she had ongoing menstrual bleeding for the past 2 1/2 weeks with an intrauterine device (IUD), a history of ovarian cysts, and PCOS. One month later, flushing and low-grade fevers had improved, but drenching night sweats continued, with increased anxiety and rage over having chronic tick-borne diseases and having been misdiagnosed. She continued seeing a therapist, following eye movement desensitization and reprocessing EMDR therapy, and we discussed brain mapping with neurofeedback for her trauma. Since she had failed clindamycin, atovaquone, azithromycin, Malarone, Coartem, tafenoquine, double-dose dapsone, lower-dose disulfiram, and antimalarial herbs for babesiosis, we discussed a trial of very low-dose mefloquine 250 mg, one-half pill per week, along with a gradual increase in disulfiram to 125 mg per day (and higher if tolerated), which has been shown to have an effect on *Babesia*, immediately stopping if there were any increase in psychiatric symptoms or neuropathy. Fluoxetine (Prozac) was tapered and changed to paroxetine (Paxil) 5 mg per day for her increased anxiety, gradually increasing to 10 mg per day.

By July 2021, she had improved to 87% of normal functioning, with less muscle and joint pain and a slight improvement in cognition, but neuropathy was still severe, with neck/shoulder/TMJ (temporal mandibular joint) pain, and she had drenching night sweats up to 4× per night. The pharmacist did not initially fill the mefloquine. She was now using ketamine twice a week, as per her psychiatrist’s instructions, since she had plateaued with Annie Hopper DNRS and Paxil. Atovaquone was DC’d, and she started a rifampin pulse 300 mg, 2 PO BID one day a week for her resistant neuropathy, while considering her first trial of HDDCT. She went back on dapsone 100 mg per day with high-dose folic acid, started mefloquine, one-half pill twice a week, began using ivermectin, and worked up her disulfiram to 250 mg PO BID for babesiosis. There were no significant Herxheimer reactions with disulfiram and no increased psychiatric effects from mefloquine. Although her sweats increased initially on mefloquine, they decreased by week 4; so, the mefloquine was stopped, but the sweats relapsed several weeks later, and mefloquine was restarted. Her primary symptoms in August 2021 included severe fatigue, pain, anxiety, light sensitivity, and night sweats. Her *Bartonella* foot pain also had increased 3 weeks prior. We discussed receiving her first round of HDDCT for resistant Lyme, *Babesia,* and *Bartonella* symptoms, and tapering down her SSRI while avoiding histamine-sensitive foods to avoid potential side effects of using methylene blue (MB), i.e., serotonin syndrome, and/or hypertension due to an MAO (monoamine oxidase inhibitor) type effect. She was placed on MegaIgG (bovine immunoglobulins, Microbiome Labs, Glenview, IL, USA) and MegaSpore probiotics (Microbiome Labs) for extra GI support for her histamine-type reactions with foods.

By October 2021, the patient had completed her first course of HDDCT with dapsone 200 mg PO BID for 4 days, along with Plaquenil 200 mg PO BID, nystatin 500,000 units 2 PO BID, minocycline 100 mg PO BID, rifampin 300 mg PO BID, disulfiram 250 mg PO BID, and methylene blue (MB) 100 mg PO BID. She noticed a “massive improvement” in symptoms, rating her percent of normal functioning at 98%. Joint pain had decreased in all of her joints, except her knees; her neck and shoulder pain was less severe (feeling like it was related to a prior whiplash injury), and pain decreased in the palms and soles of her feet to 2/10 intensity, only twice a week, although neuropathy was still present on the top of her head, one of her earliest Lyme disease and tick-borne symptoms. Her anxiety felt work-related, and ketamine infusions had significantly helped her underlying psychiatric symptoms over time. She therefore decided to do a second course of HDDCT, this time following an 8-day dapsone course with 4 days of double-dose dapsone and 4 days of high-dose dapsone with her prior antibiotics, DSF 250 mg PO BID and methylene blue. After a Herxheimer reaction, the neuropathy on the top of her head was gone by November 2021, joint pain in her knees, ankles, and wrists had decreased in frequency and almost disappeared, neck and shoulder pain was better, brain fog was gone, the pain in the palms and soles of her feet were less intense, and her bladder pressure was gone, even with methylene blue, implying a probable overlapping interstitial cystitis (IC) from Lyme disease and *Bartonella*. Adrenal support was increased in the morning and afternoon for increased fatigue by 3 PM, as a repeat salivary DHEA/cortisol test showed low cortisol levels in the morning (5.1 nmol/L, optimal range 14–25), and low cortisol levels at noon (1.3 nmol/L, optimal range 5–10). Laboratory values posttreatment in December to January 2021 were within normal limits, with a normal CBC (hemoglobin of 15.3 g/dL and hematocrit of 45.2%), normal CMP, and slightly low serum zinc (52.1 µg/dL, normal range: 60–120); so, one extra zinc glycinate 20 mg per day was prescribed. An initial methemoglobin level was elevated, at 7.6%, during treatment (normal range: 0–1.5) on dapsone and MB, but the patient denied any significant symptoms of methemoglobinemia as she denied any blue hands, blue lips, headaches, significant fatigue, or shortness of breath.

In January 2022, low-grade fevers, day sweats, and night sweats returned full-force after a booster for COVID, along with increased myalgias (mild), neck and shoulder pain, joint pain in the knees, wrists, and fingers (intermittent), light sensitivity, headaches, and brain fog and worsening cranial neuropathy. She had been feeling better up until that point, except for multiple herpes (HSV-1) flareups, requiring a course of famciclovir. Gastrointestinal issues also returned with possible *Candida* post antibiotics; so, fluconazole 200 mg QD X 14 days was prescribed with triple probiotics, followed by atovaquone, 2 teaspoons PO BID, with a high-fat meal, nystatin 500,000 units, 2 PO BID, and Bactrim DS, one PO BID, with artemisinin SOD, one PO TID, for relapsing babesiosis. When she finished that course, the plan was to discontinue Bactrim, DSF, and atovaquone, evaluate her clinical course, and remain off antibiotics, eventually restarting doxycycline, rifampin, azithromycin, and dapsone for a third pulse of HDDCT if symptoms persisted. Post HDDCT, all laboratory results were WNL (CBC, CMP) with a mildly elevated methemoglobin at 1.8% (normal range: 0.4–1.5) and ALT at 36 IU/L (normal range: 0–32). The patient received a third course of HDDCT in February 2022 (4 days of DDDCT and 4 days of high-dose dapsone, i.e., HDDCT) and remained well, feeling almost 100% of her normal level until a skiing accident 3 weeks later. The trauma caused a relapse in symptoms, although her neuropathy and fatigue had resolved for the first time in years, her mood was normal, and her cognition was also back to 100% of her normal level before the accident. It took three pulses of HDDCT to return to 100% of normal functioning, although sweats and anxiety were creeping back after the skiing accident. She felt that her *Bartonella* symptoms were temporarily in remission, and six ketamine infusions with two boosters helped significantly. The patient then had to travel to Brazil for work and contracted COVID in April 2022, resulting in moderate sweats and chills, sneezing, rhinorrhea, a mild dry cough, shortness of breath, loose stools, and a severe sore throat with chest congestion. She was prescribed a 5-day course of nirmatrelvir and ritonavir (Paxlovid), along with NAC, alpha lipoic acid, glutathione 2000 mg 2–3× per day, curcumin, Oncoplex ES, K2 D3, melatonin 1 mg HS, zinc 20 mg PO BID, and Immunotix (3, 6 beta glucan) 500 mg QD. She had received two Pfizer vaccines and two boosters to date, which she felt helped expedite her resolution of symptoms.

By June 2022, the patient was up to 98% of normal functioning, working out and traveling internationally in a new job, although moderate sweats persisted, with occasional quadricep and knee pain and pain in the bottom of her feet, lasting minutes and then resolving. The third pulse of HDDCT significantly decreased joint and muscle pain, brain fog, and light sensitivity and stabilized her mood, which was significantly better (along with EMDR). She felt that every four-day pulse of HDDCT improved her health, although mild symptoms persisted. She therefore did a fourth course of HDDCT. Her only residual symptoms post COVID were increased herpes outbreaks, responding to valacyclovir, and occasional UTIs. The last round of *Babesia* treatment flared her symptoms, and then, she temporarily recovered. She required another 14-day course of fluconazole 200 mg per day for *Candida* symptoms as she was unable to remain on a low-carbohydrate diet while traveling, remaining on her biofilm agents with essential oils (cinnamon/clove/oregano oil and Biocidin) and probiotics, which helped minimize *Candida*. Repeat laboratory testing through IgeneX laboratories revealed a CDC positive IgM Lyme Immunoblot (23, 31+/−, 41, 93), which was essentially unchanged, except for a slight increase in banding at OspC and decreased banding at Osp A. *Babesia duncani* IgM IFA went from 1:80 to <20, and *Babesia* FISH was negative. A CBC, a CMP, d-dimer, CD4/CD8 ratio profile, immunoglobulins, hemoglobin A1c, fasting insulin levels, leptin, thyroid functions, mineral levels, VEGF, and vitamin D levels were all WNL, except for an elevated IgM serum antibody (328 mg/dL, normal level 26–217), positive EBV nuclear antigen IgG with a negative Epstein-Barr PCR, and mildly elevated serum aluminum at 6 µg/L (normal range: 0–9). Despite the negative *Babesia* FISH, there was no other explanation for her ongoing sweats as pulmonary tuberculosis (TB) and a non-Hodgkin’s lymphoma (NHL) had been ruled out (negative CXRs), all hormone studies were within normal limits, and *Brucella* titers were low positive with a negative agglutination; so, she followed a longer course of tafenoquine (loading dose of 200 mg QD × 3 days, then one QD × 6 weeks) with Malarone four QD, ivermectin, *Cryptolepis sanguinolenta*, and artemisinin, which significantly improved her sweats and *Babesia* symptoms.

In August 2022, she recovered from COVID for a second time using Paxlovid with antioxidant and vitamin/mineral support (NAC, GSH, ALA, curcumin, Oncoplex ES, melatonin, vitamin D, zinc, and Immunotix), with a lingering sore throat and postnasal drip. She was placed on cetirizine (Zyrtec) 10 mg, one PO QD, with famotidine 10 mg QD, cutting back on histamine foods. D-mannose, cranberry extract, and 1 teaspoon of bioactive silver, Argentyn 23 QD (Natural Immunogenics, Sarasota, FL, USA) kept her UTIs in check. Her high-stress job, frequent travel, and some anxiety interfered with her sleep (only receiving 6 h per night), requiring extra herbal support (Copacalm with GABA/L-theanine, one to 2 dropperfuls HS, Ortho Molecular Products) while using trazodone 100 mg HS with cyclobenzaprine 5 mg HS PRN. Despite her insomnia, she nevertheless reported feeling significantly better over time, where each pulse of HDDCT relieved a resistant symptom, improving her health and resilience. She described each pulse as “peeling an onion”, removing an underlying symptom. Her neck pain had been gone for 3 months, which recently returned with joint pain in her knees and elbows post COVID. We discussed following a fifth pulse of HDDCT and receiving Gupta limbic retraining for her history of anxiety, PTSD, and trauma. By November 2022, the longer course of tafenoquine significantly helped decrease *Babesia* symptoms, and a fifth pulse of HDDCT continued to improve her health, functioning at 98% of the normal level. Her biggest improvements post HDDCT were a significant increase in energy level and cognition and decreased neuropathy; however, she began complaining about increased sinus infections, congestion, and bronchitis in August, with a return of UTIs, requiring antibiotics, which led to a relapse of candidiasis.

Due to her stress, lack of sleep, and frequent infections, a repeat adrenal test was performed, which showed phase III adrenal dysfunction with low morning and noon cortisol below range; so, extra adrenal support was instituted, using Adrenaliv, two capsules upon awakening and two capsules at noon with Adaptanall, two capsules in the morning, which helped stabilize her energy level. A mold test was also sent to RealTime Laboratories to evaluate her resistant sinus and URI symptoms, which returned positive for ochratoxins, aflatoxins, trichothecenes, gliotoxins, and zearalenone. Lyme disease and *Bartonella* symptoms also appeared to still be playing an underlying role since they were flaring up around her menstrual cycle, but otherwise, her fifth, four-day pulse of HDDCT kept the symptoms in remission most of the month. She was started on a mold detoxification protocol with phosphatidylcholine 3 g PO BID (Phosphaline 4:1, Xymogen), NAC 600 mg PO BID, glutathione 500 mg PO BID, glucomannan (Optifiber Lean, Xymogen), one scoop in between breakfast and lunch, and a charcoal/bentonite clay/zeolite binder (G.I. Detox, Biobotanical Research), one QD mid-day, two hours away from all medications and supplements. By November 2022, the patient continued to slowly improve, but her last bout of COVID led to an increase in night sweats, *Candida* issues, and viral relapses with herpes outbreaks, and she continued to complain about insomnia secondary to underlying anxiety and job stress. A two-week pulse of fluconazole with berberine and Biocidin reduced *Candida* symptoms, and she was treated with a repeat 6-week course of tafenoquine (loading dose 200 mg QD × 3 days, then 200 mg Q week), ivermectin (based on body weight), and Malarone, 4 QD, since the last round of treatment helped decrease *Babesia* symptoms. Trazodone 50–100 mg HS, cyclobenzaprine 5 mg HS, and herbal support (Copacalm) eventually improved her resistant insomnia.

Between August and December 2022, the patient contracted COVID for a third and fourth time, with influenza A six weeks prior, resulting in increased fatigue, rhinorrhea, a sore throat, and chest congestion. Oseltamivir (Tamiflu) 75 mg PO BID was prescribed ×5 days, and Paxlovid had to be repeated during one course of treatment secondary to a relapse in symptoms post therapy. We discussed rechecking immune function with immunoglobulins and subclasses, natural killer cells (total, function), and mineral and vitamin levels, along with pre- and post-pneumococcal titers. No etiology could be found for her frequent infections, apart from her history of low adrenal function and stress, chronic insomnia, and mold toxins. The patient noted increased herpes flareups post COVID despite being on Valtrex, which were associated with a temporary increase in sore throats, joint pain, sweats (consistent with a *Babesia* relapse), and intermittent sore throats. Transfer Factor Multi-immune (Researched Nutritionals) was added to 3,6 Beta glucan, zinc, and glutathione for enhanced natural killer cell/immune support. A repeat course of Coartem, three days on and 11 days off, was prescribed with ivermectin, Malarone, and *cryptolepis sanguinolenta*, 1 teaspoon TID, for her sweats since she was leaving for South Africa in one month. She felt this *Babesia* treatment was more helpful than her last course of tafenoquine. We discussed receiving a sixth course of HDDCT if the symptoms persisted, gradually adding on minocycline, rifampin, azithromycin, and pyrazinamide in week one and extending the course of dapsone to 6–7 days of dapsone 200 mg PO BID in the last week, along with a higher dose of methylene blue at 300 mg PO BID. The patient would taper off her trazodone and cyclobenzaprine prior to treatment to avoid interactions with methylene blue (MB), remaining on a low histamine diet.

By February 2023, the night sweats had improved, but her worst symptoms included moderate fatigue, resistant insomnia (4 h of solid sleep per night), weight gain, internal tremors, twitching, myalgias, joint pains in the knees and elbows, with pain in the soles of the feet, intermittent neuropathy (legs, occasionally hands, and scalp), headaches, light and sound sensitivity, and brain fog. She felt that her continued unresolved trauma was affecting her symptoms (although EMDR, Gupta limbic retraining, and ketamine infusions significantly helped) [97], along with contracting COVID two more times (6× in total) [98]. She went for periods of time where she was relatively asymptomatic in between dapsone pulses, and her symptoms with neuropathy would only flare up temporarily post COVID and viral relapses. The prior five pulses of HDDCT had removed layers of her illness, improving with each pulse, and her longest remission with *Bartonella* symptoms was four months. We therefore prescribed a sixth pulse of HDDCT using Pyridium (phenazopyridine) PRN for urinary discomfort on high-dose methylene blue, 300 mg PO BID. Her prior HDDCT pulses were four days in length, with lower doses of MB. This was her first 14-day pulse of antibiotics, using a high-dose dapsone pulse for the last seven days, with Plaquenil, nystatin, minocycline, rifampin, azithromycin, pyrazinamide, and a higher dose of MB (300 mg PO BID) with antioxidant/nutritional support.

As of January 2024, the patient’s tick-borne symptoms were in full remission for 12 months or longer after the longer pulse of high-dose dapsone combination therapy. She denied any fevers, fatigue, pain, headaches, neuropathy, anxiety, light and sound sensitivity, or brain fog. She felt that the stress of her new job would occasionally cause herpes “tingles” and that her sweats would only come out with trauma therapy but that otherwise she was in full remission since no further Lyme/*Babesia*/*Bartonella* symptoms were relapsing during her menstrual cycle, which previously indicated an ongoing infectious load. A summary table of co-infection status, MSIDS variables, and the number of DDDCT and HDDCT pulses required to achieve remission in all three cases can be found in Table 1.

## 3. Minimizing Potential Side Effects of Dapsone Combination Therapy/Prescribing Information

The potential side effects of HDDCT involve four basic categories of symptoms that can be summarized as “Do No H.A.R.M.”, where H stands for Herxheimer reactions, A for anemia (secondary to folic acid inhibition and potential hemolysis), R for rashes (if sulfa sensitivity is present), and M for methemoglobinemia (decreased oxygen-carrying capacity). These were addressed using medication and nutritional supplements/antioxidants with anti-inflammatory effects, including glutathione, N-acetylcysteine, and alpha lipoic acid, which block NFKappaB [99], and turmeric and broccoli seed extracts (sulforaphane glucosinolate), which stimulate Nrf2 [100]. Glutathione, nicotinamide adenine dinucleotide with hydrogen (NADH), vitamin C, vitamin E, and methylene blue were all used to decrease methemoglobin levels [101,102,103], along with the occasional use of cimetidine to lower methemoglobin in those with a history of significant methemoglobinemia [96,104], while high-dose folic acid (folinic acid, i.e., leucovorin and L-methyl-folate) was prescribed to minimize anemia from the inhibition of bacterial synthesis of dihydrofolic acid by dapsone [105]. In prior studies, folinic acid supplementation has been shown to help limit myelosuppression, gastrointestinal toxicity, nephrotoxicity, and neurotoxicity, which can result from high dosages of folic acid antagonists [106]. High-dose probiotics including *Saccharomyces boulardii* were used to help prevent antibiotic-associated diarrhea [106,107].

In order to minimize potential interactions with methylene blue, including serotonin syndrome/toxicity [108], patients on psychiatric medications, especially a selective serotonin reuptake inhibitor (SSRI), monoamine oxidase inhibitor (MAOI), and/or bupropion (Wellbutrin), were told to taper their psychiatric medication if clinically stable and/or consult with their psychiatrist before increasing the dose of methylene blue to a maximum final dose of 300 mg twice a day. Dietary restrictions also were instituted, avoiding foods high in tyramine, which can potentially produce a hypertensive crisis in the presence of MAOIs and/or higher-dose methylene blue [109]. Patients were given a list of foods to avoid during the protocol, which included aged cheese, aged chicken or beef liver, air-dried sausage and similar meats, avocados, beer, wine (in particular, red wine), canned figs, caviar, fava beans, meat tenderizer, overripe fruit, pickled or cured meat or fish, raisins, sauerkraut, shrimp paste, sour cream, soy sauce, and yeast extracts [110]. Phenazopyridine (Pyridium) was used at a dose of 200 mg TID prn if any urinary discomfort/burning arose from the use of higher-dose methylene blue [111].

In our prior study [39], several patients took disulfiram (DSF) in combination with HDDCT if they had previously failed to have adequate clinical improvement with either drug regimen used alone or in combination, utilizing higher dosing if they failed prior combination therapies. The patients signed a consent informing them of the potential benefits and risks of DSF, including increased fatigue, brain fog/cognitive dysfunction, worsening psychiatric symptoms, liver function abnormalities, and/or increased neuropathy [112,113]. They were instructed to stop DSF immediately if there was any significant worsening in underlying symptomatology, especially neuropathic symptoms [114], and to contact our office immediately if any severe adverse effects were noted. The dose of DSF that was used ranged from a lower-dose DSF (250 mg or less) to a higher-dose DSF (500 mg/day). The majority of patients who took DSF in our study used doses of 250 mg/day or less to minimize the potential adverse effects and Herxheimer reactions. The disulfiram dose was slowly increased over time, increasing the dose by 62.5 mg every one to two weeks until reaching the target dose, which was based on efficacy and tolerance. The patients were instructed to use sodium bicarbonate or freshly squeezed lemons and/or limes to alkalize the body if Herxheimer reactions resulted from the killing off of *Borrelia* [115], along with N-acetyl cysteine, alpha lipoic acid, and high-dose glutathione (2000 mg). These nutraceuticals can help to decrease inflammation through their blocking effect on NFKappa-B, lowering inflammatory cytokine production. Apart from the above nutritional supplements and dietary recommendations, the DSF group was also instructed to avoid ingestion and exposure to alcohol to minimize gastrointestinal complications, including nausea and/or vomiting, known side effects of DSF toxicity [116]. All patients on tetracyclines were instructed to avoid direct sun exposure for more than several minutes to avoid the possibility of sunburn, use a sun protection factor of 65 or above, take the medication with a full 8-oz. glass of water after a meal, sitting up for one hour to avoid any reflux esophagitis, and avoid concomitant use of dairy products and minerals within one hour of the tetracycline to avoid decreasing the absorption of the medication [117].

Although both dapsone and disulfiram have demonstrated some efficacy against resistant biofilm/persister forms of *Borrelia burgdorferi* [71,118,119], up to four different biofilm agents were regularly used during the DDDCT and HDDCT protocols to improve efficacy. These included Stevia [120], Biocidin [121], essential oils, including oregano, cinnamon, and clove [122], and peppermint oil [123]. Rarely, a fifth biofilm agent, namely ethylenediaminetetraacetic acid (EDTA) suppositories (Detoxamin, 750 mg) [124], was considered in the last week of HDDCT if patients had extremely severe symptoms and/or had failed prior regimens with DDDCT and HDDCT, with or without bioactive silver hydrosol (Argentyn 23), which was used for its antibacterial and synergistic effects [125]. If patients were on DSF, monolaurin and serrapeptase were substituted as biofilm agents instead of using Stevia and Biocidin since they do not contain any alcohol [126,127]. Grapefruit seed extract and hydroxychloroquine were also used for their effects against the round body forms of *Borrelia* [128,129].

The medication and nutritional supplementation protocol sheet for double-dose dapsone combination therapy (DDDCT) and high-dose pulsed dapsone combination therapy (HDDCT) for CLD/PTLDS outlined in detail the protocol that our three patients needed to follow closely and is in our prior publication in Microorganisms, September 2023 [39]. This should be referred to for further reference. Some essential points from that publication include using a low histamine diet to protect against significant increases in blood pressure while on higher-dose methylene blue (MB), avoiding certain psychiatric drugs on MB due to the potential for serotonin syndrome, staying on a low-carbohydrate diet and using up to four different probiotics to avoid promoting yeast/candida overgrowth while on antibiotics, and ensuring that the patients were glucose-6-phosphate dehydrogenase (G6PD)-positive without significant iron deficiency or vitamin deficiencies (B12 and folate) before starting the protocol to avoid worsening anemia. Any medication interactions, due to potential interactions with rifampin, cimetidine, macrolides, ondansetron, etc., were also accounted for, as well as checking an EKG and QT intervals before and during the protocol when several QT-prolonging medications were used (macrolides, hydroxychloroquine, and ondansetron). Severe Herxheimer reactions [115] were addressed using 2 g of glutathione all at once, up to three times daily until the Herxheimer reaction resolved [30,130], with up to 2000 mg of glutathione 3× per day if symptoms of methemoglobinemia [94] (blue hands, blue lips, headaches, fatigue, and shortness of breath) occurred.

After finishing the DDDCT and HDDCT protocols, a one-month mitochondrial regeneration program was given to all three patients, which was also given to all 25 patients in our initial publication: ATP 360, three capsules once a day (Researched Nutritionals), ENADA (NADH), one a day, carnitex (Xymogen), two twice a day (not for those with alpha-gal allergy), CoQ Power 400 mg twice a day (Researched Nutritionals), Cardio Ribose (Researched Nutritionals), one scoop twice a day, along with Mitoprime (Xymogen), one a day, and Mito NR (Designs for Health, Suffield, Connecticut), two a day. Depending on the patient’s symptoms, underlying conditions, and responses to the protocol, a cost–benefit analysis of finishing each pulse with mitochondrial regeneration should be performed.

## 4. Discussion

The etiology of symptoms associated with chronic Lyme disease (CLD)/PTLDS has been a highly debated topic in the scientific literature since the discovery of *Borrelia burgdorferi* in 1981 [131], where researchers and clinicians with opposing views have published evidence-based clinical practice guidelines for the prevention, diagnosis, and treatment of Lyme disease [3,132]. Both guidelines agree, however, on certain essential points in the treatment of patients. First, the International Lyme and Associated Diseases Society (ILADS) guidelines state that “different choices may be appropriate for different patients…and that shared decision making to engage in a risk–benefit assessment that reflects the individual values of the particular patient” is important [3], while the Infectious Disease Society of America (IDSA) guidelines state that “guidelines cannot always account for individual variation among patients… may not reflect the most recent evidence… should not be considered inclusive of all proper treatments methods of care… and are not intended to supplant physician judgment with respect to particular patients or special clinical situations” [132]. Personalized, precision medicine approaches are thereby implied in both sets of guidelines, taking into account new advances in science, where it is important to consider environmental, socio-economic, psychological, and biological determinants [133], as well as practical applications and bioethics [134]. Two publications outlining the importance of precision medicine approaches in 200 chronically ill Lyme disease patients were published by Horowitz et al. in 2018 and 2019 [14,30], highlighting the important role of overlapping sources of inflammation and downstream effects in the diagnosis and treatment of CLD/PTLDS, and since the publication of the last ILADS guidelines in 2014 and the 2020 IDSA guidelines, there have been several new clinical studies outlining novel effective treatments.

Pothineni et al. published that disulfiram, a Federal Drug Administration (FDA)-repurposed medication, can inhibit *Borrelia* in culture [120], while Liegner et al. demonstrated that both low- and especially high-dose disulfiram regimens were able to put some chronically ill Lyme patients into long-term clinical remission [114,115]. From 2016 to 2023, Horowitz et al. published eight studies on the efficacy of dapsone, another FDA-approved repurposed medication, where dapsone was shown to be effective in culture against the biofilm/stationary forms of *Borrelia burgdorferi* [75] and where over 375 patients had statistical improvement in their Lyme disease and tick-borne symptoms using dapsone combination therapy and the precision medicine MSIDS model [14,30,38,39,71,72,73]. An evidence-based definition by the ILADS Working Group in 2019 further clarified the term ‘chronic Lyme disease’ (CLD), providing a literature review that validated the role of persistent *Borrelia burgdorferi sensu lato* (*Bbsl*) infections in chronic disease [135]. Previous studies in mice [136], horses [137], dogs [138,139], and macaques [140,141] proved that *Borrelia* can establish a chronic, persistent infection post antibiotic therapy, and human studies also showed that *Borrelia* can persist in the eyes [142,143], central nervous system [144], ligaments [145], joints [146], intracellular compartments [147,148], macrophages [149], fibroblasts [150], as well as in the cutaneous tissue with biofilm formation [151]. Immune evasion by evading the complement system [152] and resistance to antimicrobial peptides [153] while exhibiting pleomorphism [154] and changing forms in different environmental conditions may also contribute to its ability to establish a persistent infection in the host [155].

Biofilm formation by *Borrelia burgdorferi sensu stricto* and *Bb sensu lato*, both in culture [83,156] and in human studies [151,157], and the ability of the bacteria to form inactive stationary forms/persisters in biofilm-like microcolonies, increasing inflammation while contributing to antibiotic resistance [158], appear to be important mechanisms causing persistent symptoms in CLD/PTLDS based on the success of dapsone combination therapy in eight published studies. In the three case studies presented here, all three patients had failed to achieve clinical remission with single-drug antibiotic therapy or combination antibiotic therapy, whether it was PO, IM, or IV, during months or years of administration, until adequate doses of persister drug regimens with biofilm agents were administered and/or eventually pulsed. In case number 1, cefdinir 300 mg PO BID, grapefruit seed extract (GSE), clarithromycin 500 mg PO BID, sulfamethoxazole/trimethoprim (Bactrim DS), and atovaquone/proguanil (Malarone) were given for two years without antibiotic rotations. Although the patient reported an improvement from 5% of normal functioning to 65% of normal functioning after one month of treatment, she then plateaued without further improvement during the next 23 months, with flareups of significant flulike symptoms, fatigue, pain, and brain fog every three weeks. Once the patient was placed on dapsone combination therapy, using two persister drugs, rifampin, and dapsone, even low-dose dapsone (25 mg QOD) with doxycycline 100 mg PO BID, rifampin 300 mg PO BID, and Malarone two PO BID led to an increase in energy levels, decreased joint and muscle pain, decreased neuropathy and twitching, increased cognitive functioning, and an improvement in *Babesia* symptoms. After following several months of higher-dose dapsone, 50–75 mg, and one month of 100 mg of dapsone, by December 2018, 5 months off all antibiotics, she remained at 95% of normal functioning, and there were no clear menstrual flares with a relapse of underlying Lyme disease and tick-borne symptoms. Dapsone combination therapy helped her to remain in remission for seven months until February 2019, and even then, she was not sick enough to go back on antibiotics until April 2022, three years later, several months post COVID, when tick-borne symptoms relapsed. Post COVID, there was a return of night sweats several times per month, increased motion sickness, moderate fatigue, and joint pain in the arms, legs, and hips several times per week (which would flare up during her menstrual cycle), along with word-finding issues and moderate difficulty thinking. Although she was significantly better than years ago with lower dose dapsone combination therapy (maximum dose 100 mg), because she was going to college in a year, we rechecked her tick-borne testing, and her *Bartonella* FISH returned positive. The patient did the full 9-week oral dapsone combination therapy with one month of double-dose dapsone (100 mg BID) and 6 days of high-dose dapsone for *Bartonella* for the first time in her illness and remained in full remission without any further return of TBD symptoms for over one year. She still, however, required addressing MSIDS variables for her to remain well. These include treating her POTS, CVID, phase 2 adrenal dysfunction with metabolic syndrome/reactive hypoglycemia, and mold toxicity; yet, it was not until she followed the higher-dose dapsone protocol with a tetracycline, rifampin, azithromycin, pyrazinamide, and higher-dose methylene blue that her Lyme disease and *Bartonella* symptoms remained in full remission, feeling 100% back to normal.

In case number 2, she also failed to achieve full remission for her TBD using single-drug antibiotic therapy or combination antibiotic therapy (PO, IM, or IV) during months or years of administration until adequate doses of persister drug regimens with biofilm agents were administered and eventually pulsed. Her first Lyme disease protocol included doxycycline and atovaquone/proguanil (Malarone), which was switched to cefuroxime axetil, resulting in severe confusion, fatigue, and increased arthralgias. Over time, her condition worsened. She was then given intramuscular (IM) ceftriaxone (Rocephin) twice weekly with minocycline 100 mg twice a day, which improved her cognition and other symptoms but was switched to IV ceftriaxone because of an inadequate response. She subsequently saw another physician who treated her for Lyme disease and *Bartonella*, where she took different cocktails of antibiotics during the next several years, including hydroxychloroquine (Plaquenil), rifampin, azithromycin, pyrazinamide, and methylene blue for two positive T Lab *Bartonella* FISH tests. Due to gastrointestinal (GI) upset and lack of adequate efficacity, she was rotated over to IV medications and received nine months of IV rifampin, IV azithromycin, and pulse IV ceftriaxone, along with eight months of disulfiram. Despite these long-term oral and IV regimens, which used some persister medications, she remained chronically ill with fatigue, pain, and resistant psychiatric symptoms (hallucinations). One of the last pharmaceutical regimens, which she had tried before coming to our medical office, was IV gentamicin, along with 16 months of oral and IV treatment for Lyme disease and *Bartonella*. Yet, despite these protocols, she still tested positive for active *Bartonella* by a FISH test through T Lab, although there had been some clinical improvements. At the time of her presentation to our medical office, despite years of ongoing oral and IV medication, her symptoms were relapsing off antibiotics during the prior three months, including moderate fatigue, moderate joint pain in the hips and knees that were migratory, neuropathy of the lower extremities with foot pain, moderate cognitive difficulties, anxiety, a movement disorder, and ongoing hallucinations. The patient had been on dapsone for two months, up to 100 mg per day, prior to seeing us, but had never been on dapsone 200 mg (i.e., double-dose dapsone) or pulsed high-dose dapsone (400 mg per day for 4–6 days) with higher dose methylene blue for her chronic Lyme disease and *Bartonella*. In her case, simply taking persister drugs with disulfiram and dapsone, even in combination therapy, was insufficient to lead to a clinical remission. Based on published medical research, the doses she used would have been inadequate to control and effectively eliminate the infections [38,39,73].

It was not until she completed an eight-week course of double-dose dapsone, i.e., DDDCT (200 mg PO BID), four 2-week pulses of high-dose dapsone (HDDCT) x 4 days, with higher-dose methylene blue, and two 6-day pulses of HDDCT that she eventually had a full remission of symptoms. She reported that each dapsone pulse led to longer periods where she remained asymptomatic, and clinically, it appeared as if the load of the bacteria decreased with each pulse of dapsone combination therapy. However, it was not until having received several 6-day pulses of HDDCT, which brought out several underlying symptoms that had been in remission, that she felt 98% back to normal. She would have listed her improvement at 100% but had one hour of brain fog after going to the dentist one day prior and one day of fatigue while traveling during her menses (which her sisters also experience around their menstrual cycle). There were, however, no further Herxheimer flares around her menses after her second 6-day HDDCT pulse, which had previously been an issue with prior treatments. At this point in time, the patient was approaching two months in full remission, without any regular fatigue, pain, neuropathy, movement disorders, or neuropsychiatric symptoms, stating that “she is feeling great and better than she has ever felt!”, although it is possible that she will require one or more two-week pulses, i.e., 6 days of HDDCT, in the future, should symptoms relapse. Among the 25 patients in our initial dapsone publication, comparing longer vs. shorter pulses, those who completed at least one 6-day HDDCT pulse had a three-month remission rate of 30.4%, and those who had received multiple 4-day HDDCT pulses with one or more 6-day HDDCT pulses had more enduring remission, such as case number 3.

The superiority of 6-day HDDCT pulses for *Bartonella* was highlighted in another of the 25 chronically ill patients who were reported in our initial publication on the benefits of longer high-dose dapsone pulses [39]. Patient number 20 in that study had chronic Lyme disease and *Bartonella* and was in remission for one year after double-dose dapsone combination therapy, but when given antibiotics for an unrelated infection (cellulitis), the patient had a relapse of symptoms one year later, with the emergence of new *Bartonella* striae [39]. That patient had never received high-dose dapsone combination therapy. *Bartonella* is a known stealth pathogen that can hide in the intracellular compartment and periodically seed into the bloodstream [159], and its persistence may be underestimated due to the broad range of species and limitations of present diagnostic testing [160]. Case number 2, highlighted here, who is now in remission, had neuropsychiatric symptoms along with classic *Bartonella* striae. These skin lesions, resembling ‘stretch marks’, which can be horizontal or perpendicular to the skin planes, whitish or purplish in color, are now being reported more frequently in the medical literature in those with neuropsychiatric presentations and, therefore, can be an important clinical sign pointing to the presence of *Bartonella* spp. [161,162]. *Bartonella* striae returned in case number 2, right before receiving her first 6-day high-dose dapsone pulse, confirming an active infection, despite four 4-day pulses of HDDCT, which were inadequate to control the infection. Each 4-day HDDCT pulse led to longer periods without symptoms, but two 6-day pulses of HDDCT were needed for greater improvement. The superiority of 6 days of combination antibiotic therapy using persister medication, i.e., rifampin and methylene blue with azithromycin and biofilm agents, is consistent with Johns Hopkins studies on the efficacy of these medications in eliminating *Bartonella* in culture. In a study by Zheng et al., methylene blue and rifampin were the most active agents against the biofilm *B. henselae* after 6 days of drug exposure, and azithromycin/methylene blue, as well as rifampin/methylene blue, completely eradicated the biofilm aggregates of *B. henselae* after treatment for 6 days [76].

Case number 3 highlighted the presence of striae and overlap of *Bartonella* spp. with severe neuropsychiatric symptoms, the importance of addressing abnormalities on the 16-point MSIDS map, as well as the superiority of 6-day HDDCT pulses for the treatment of CLD and *Bartonella*. Her symptoms started in 2008 when she was living in the woods in Germany, in a highly endemic area for Lyme disease [163]. She had striae, which developed on her flanks, buttocks, and inner thighs, and had cats growing up, receiving multiple bites and scratches; yet, she had never been diagnosed with *Bartonella* [164]. Despite increasing fatigue, migratory myalgias, migratory neuropathy, cognitive difficulties, and severe neuropsychiatric symptoms, it took four years for the patient to be diagnosed with chronic Lyme disease, relapsing fever, *Babesia duncani*, and anaplasmosis, although migratory muscle and joint pain, and especially migratory nerve pain, is a characteristic clinical symptom of chronic Lyme disease [165]; relapsing and remitting night sweats in a young woman is indicative of a possible infection with *Babesia* spp. [166], and significant pain in the soles of her feet is a well-established clinical sign of exposure to *Bartonella* [167]. She was treated with 6 months of nitazoxanide (Alinia) for *H. pylori* after following one week of doxycycline, which was then rotated to amoxicillin with clarithromycin and omeprazole for approximately one year, without any improvement. When she came to see us, overlapping MSIDS variables contributing to her chronic illness included reactive hypoglycemia, chronic candidiasis, possible POTS/dysautonomia due to her extensive neuropathy and dizziness while standing with syncopal episodes, depression, anxiety, premenstrual dysphoric disorder (PMDD), and post-traumatic stress disorder (PTSD) secondary to trauma and abuse, chronic insomnia with restless leg syndrome, chronic UTIs with possible interstitial cystitis (IC) secondary to *Borrelia* [168] and/or *Bartonella* [169], intermittent outbreaks of HSV-1, HSV-2, and HPV, bacterial vaginosis, possible mold toxicity with mast cell activation and migraines, heavy metal toxicity with aluminum and mercury (tinnitus), phase 3 adrenal dysfunction with low cortisol levels in the morning, at noon, and in the evening, allergic rhinitis with exercise-induced asthma, vitamin deficiencies (B and D), mineral deficiencies (low iodine, zinc, and iron, with low ferritin), low glutathione levels, and possible mitochondrial dysfunction. Many of these overlapping factors on the MSIDS map were increasing underlying symptomatology, apart from her tick-borne infections since hypoglycemia, *Candida*, POTS, low adrenal function, chronic insomnia, depression, low glutathione levels, vitamin and mineral deficiencies, heavy metal toxicity, and possible mitochondrial dysfunction can all increase underlying fatigue and affect mood [170,171,172,173,174,175,176,177,178,179]. Similarly, infections with *Borrelia*, *Bartonella*, and *Anaplasma*, heavy metal toxicity, i.e., mercury (Hg), lead (Pb), and arsenic (As), mold toxicity, vitamin B deficiency, and mitochondrial dysfunction can all cause neurological symptoms and increase neuropathy [180,181,182,183,184,185,186], which is why a full MSIDS evaluation in CLD/PTLDS is highly important for diagnosing and treating all potential overlapping etiologies. A laboratory clue that *Bartonella* was also playing a significant role in her chronic illness was that she had an indeterminate *Bartonella* Western blot and evidence of prior exposure to *Bartonella vinsonii* sub spp. while her tularemia and *Brucella* titers returned low-level positive at 1:20, consistent with possible reactivity with other intracellular infections (like *Bartonella*) [104,105]. Chronic intravascular infection with *B. henselae* and *B. vinsonii subsp. berkhoffii* in immunocompetent people with animal contact (cats) and arthropod exposure has previously been described in the scientific literature [181,187].

The patient was initially rotated through lower-dose dapsone combination therapy for her Lyme disease, *Babesia* and *Bartonella* infections, secondary to a history of GI intolerance to medication and severe Herxheimer reactions. She had already followed a course of atovaquone and azithromycin for babesiosis, which was ineffective in relieving her sweats and malarial-type symptoms. *Babesia* had now become resistant to standard antibiotic/antimalarial regimens, including clindamycin, azithromycin, and atovaquone [188,189]; so, other antimalarial protocols were tried. Her first regimen, therefore, included minocycline and rifampin, gradual increases in the dose of dapsone, atovaquone/proguanil (Malarone), and multiple 3-day pulses of lumefantrine/artemether (Coartem) with *Cryptolepis sanguinolenta* and artemisinin for her ongoing, resistant night sweats secondary to *Babesia*, which eventually improved [190]. By November 2020, upon reaching a dose of dapsone 100 mg per day for 3 weeks with Plaquenil, nystatin, minocycline, and rifampin, she noticed significant improvements in her fevers, sweats, chills (*Babesia*), anxiety, depression, and gastrointestinal issues (no diarrhea or vomiting with GERD), improved energy levels, fewer headaches, fewer tremors, and improved sleep. Since the patient was clinically stable, with a normal hemoglobin level (13.1 g/dL), CMP, and methemoglobin level (0.00), she received her first course of DDDCT, increasing dapsone to 100 mg PO BID, with Plaquenil, nystatin, minocycline, and rifampin, while increasing her folic acid and adding methylene blue (MB) 50 mg PO BID, tapering off her selective serotonin reuptake inhibitor (SSRI) to avoid potential interactions (serotonin syndrome) with MB. Post DDDCT, compared to 6 months prior, she improved. She had less fatigue, swollen lymph nodes, myalgias, light and sound sensitivity, GERD, bladder pressure, improved cognition, and fewer mood swings (improved anxiety and depression), but she still complained about fevers, drenching night sweats HS, which were waking her up, secondary to resistant *Babesia* symptoms that returned, restless leg syndrome (RLS), headaches, worsening joint pain in the wrists, hips, ankles, and fingers, pain in the soles of her feet (consistent with active *Bartonella*), tremors, and unbearable neuropathy, despite using gabapentin 300 mg PO TID.

At this point, she had failed *Babesia* treatment with atovaquone and Zithromax, Coartem, Malarone, dapsone, and antimalarial herbs but had never used tafenoquine. She was therefore prescribed a course of tafenoquine for resistant *Babesia* [191], but unfortunately, she still complained about ongoing flushing and night sweats post treatment, and then, many underlying tick-borne symptoms worsened again after being exposed to COVID-19, including her sweats, chest tightness, fatigue, joint pain, brain fog, headaches, and neuropathy. Exposure to *Borrelia burgdorferi* has been identified as a risk factor for more severe disease with COVID-19 [24], just as *Babesia* can increase the severity of underlying tick-borne symptoms [192]. Although tafenoquine has been reported to potentially be a more effective treatment for babesiosis, based on animal studies and limited case reports [191,193], larger well-controlled studies need to be performed to evaluate its efficacy. A recent 2024 study by Yale researchers reported that the combination of tafenoquine and atovaquone has superior efficacy against *Babesia* parasites [194], but our experience with tafenoquine is that the treatment decreased symptoms and clinically appeared to lower the parasitic load (less fevers, sweats, chills, flushing, air hunger, and cough), although the symptoms often relapsed when treatment was stopped [39]. The patient did not, however, take tafenoquine with atovaquone as the study on the combination drug therapy was published one year after the patient finished treatment.

In our initial study involving 25 patients comparing the efficacy of a 6-day course of HDDCT versus a 4-day course, *Babesia* was found in 64% (16/25) of our patients, involving multiple species, including *Babesia microti*, *Babesia duncani*, *Babesia odocoilei*, and *Babesia divergens*. Among several clinical studies by Horowitz et al., involving over 350 patients with overlapping tick-borne infections, including *Borrelia, Babesia*, and *Bartonella,* where one or more species of *Babesia* were present, dapsone had some anti-malarial effect, but rotation of different anti-*Babesia* medications and combination therapies were required to achieve clinical remission [14,38,39,71,73]. The same scenario was seen in case number 3. Eventually, over time, the patient resolved her resistant *Babesia* symptoms, but it required several rotations of anti-malarial medications over a two-year period, apart from tafenoquine, including clindamycin, atovaquone, atovaquone/proguanil, disulfiram, higher-dose dapsone pulses, sulfamethoxazole/trimethoprim with azithromycin, lumefantrine/artemether, ivermectin, as well as anti-malarial herbs, including *Cryptolepis sanguinolenta*, artemisinin, Japanese knotweed, and Chinese skullcap. Further research is needed to improve the efficacy of *Babesia* treatments, including clinical studies using fosinopril, an angiotensin-converting enzyme inhibitor (ACE), which has recently been identified as a potent anti-parasitic drug with efficacy in vitro against *Babesia duncani* [195]. Another approach that deserves further study, is based on the metabolomic profiling of *B. duncani*, which identified parasite-mediated modulation of RBC metabolite levels of all classes, including lipids, where the interruption of cholesterol scavenging from the host cell led to premature parasite egress, and chemical targeting of the hydrolysis of acyl glycerides led to the buildup of malformed parasites [196]. The use of orlistat, a potent inhibitor of the lipase that degrades acylglycerides, has been shown to disrupt *B. divergens* merozoite membrane formation in culture, inhibiting merozoite maturation before egress [196]. No clinical studies have been performed to date using orlistat for the treatment of *B. divergens*, which is an FDA-approved drug for the treatment of obesity [197]. The efficacy of the drug may be limited, however, by its minimal absorption and side effects related to fat malabsorption, including abdominal pain, flatus with discharge, and fatty/oily stool [198].

In case number 3, while addressing resistant *Babesia* symptoms with anti-malarial drug rotations, between October 2021 and January 2023, the patient received a total of five 2-week pulses of HDDCT × 4 days, and each high-dose dapsone pulse relieved a layer of resistant symptoms that did not return. It was not, however, until she did a sixth dapsone pulse of 6 days for the first time in February 2023 that she managed to remain one year in full remission, without any return of her underlying fatigue, resistant insomnia, weight gain, internal tremors, twitching, myalgias, joint pains with pain in the soles of the feet, intermittent neuropathy, headaches, light and sound sensitivity, and brain fog. This was despite contracting COVID-19 six times during a period of three years due to frequent international travel on planes. Previously, she would be relatively asymptomatic in between 4-day HDDCT pulses, the longest period being 4 months, similar to case number 2, and the patient reported that her symptoms (especially neuropathy) would only flare up temporarily post COVID and with viral relapses of her herpes viruses (HSV1 and HSV2). This required the regular use of famciclovir to keep herpes flare-ups in remission [199]. This patient had been initially vaccinated against COVID-19 with frequent boosters, treated with Paxlovid (nirmatrelvir and ritonavir) during each exposure, and took high-dose glutathione and antioxidants with immune support during each infection [200]. As per the study by Cameron et al., using a General Symptom Questionnaire 30 (GSQ 30) to evaluate the burden of LD symptoms among 889 Lyme disease patients who were vaccinated or not vaccinated against COVID-19, those who were vaccinated had significantly lower neurological symptoms [201]. This included less neuropathy (numbness, tingling, shooting, stabbing, or burning pains), balance problems, visual or sound sensitivity, concentration problems, twitching, bladder discomfort or change in urination, shortness of breath, and/or irregular or rapid heartbeat. Although anecdotal reports suggested that nirmatrelvir may improve long COVID symptoms [202,203], including a Veterans Affairs (VA) study among 281, 793 patients [204], a 2024 study in the Journal of the American Medical Association (JAMA) reported no significant benefit using Paxlovid in diminishing the effects of long COVID or in decreasing the intensity of post-COVID condition in subjects vaccinated before or after infection [205]. It is therefore possible that the concomitant use of high-dose glutathione, NAC, antioxidants, and immune support used by patient 3, along with vaccination and boosters, helped mitigate long-term side effects [101,102,202] since the patient never developed symptoms of long COVID despite exposure six times and concomitant infections with overlapping medical problems. Low glutathione levels and associated thiols, including low levels of NAC, have now been published in multiple scientific studies and reviews as being associated with the most severe complications of COVID-19 infection [206,207,208]. Dapsone has also been published as having significant benefits in a small control group of COVID-19 patients with acute respiratory distress syndrome (ARDS), decreasing mortality in COVID-19 patients admitted to the intensive care unit (ICU) [209,210]. Potential mechanisms for dapsone’s efficacy in COVID-19 include decreasing inflammation via the inhibition of myeloperoxidase [211], inhibiting neutrophil tissue destruction, and acting as an NLPP3 inflammasome inhibitor [212].

Regarding dapsone’s efficacy in Lyme disease, dapsone, alone and in combination with rifampin, a tetracycline, and/or a macrolide (azithromycin), has been shown to be effective in decreasing the biofilm/persister aggregates of *Borrelia burgdorferi* in culture [75]. Each addition of another antibiotic to dapsone (doxycycline, rifampin, and/or azithromycin) increased the efficacy and ability of the antibiotic combination to lower the biofilm load of the bacteria [75]. The efficacy of dapsone combination therapy in culture against the biofilm/stationary phase aggregates of Bb was also confirmed by Johns Hopkins researchers [213]. In that study, combination antibiotic regimens with sulfa drugs combined with other antibiotics were more active than their respective single drugs, and similar to the study by Horowitz et al. [75], four-drug combinations were more active than three-drug combinations. The four-drug combination regimen of dapsone + rifampin + minocycline + cefuroxime showed the best activity against stationary-phase *B. burgdorferi* [213]. Recently, Tulane researchers confirmed the superior efficacy of dapsone combination antibiotic therapy versus monotherapy in a mouse model of Lyme disease [214]. In that study, dapsone combination therapy, using rifampin and dapsone, was effective in curing a chronic *Borrelia burgdorferi* infection in mice. This is the first animal model proven to confirm the efficacy of dapsone combination therapy in clearing a chronic infection with *Borrelia* [214]. Part of the success of dapsone combination therapy may be due to dapsone’s excellent penetration into the central nervous system, antimalarial effects, ability to lower autoimmune reactions and inflammation, and lower biofilm-like aggregates of *Borrelia burgdorferi* in culture [215,216]. Our patient, who was positive for human leukocyte antigens (HLA) DR2/4, which are independent dominant markers of susceptibility and treatment-refractory Lyme arthritis [217], also resolved her resistant arthritic symptoms with dapsone combination therapy. Earlier studies by Johns Hopkins researchers showed that these biofilm aggregates were a major source of inflammation [158], and inflammation, whether from chronic infection or overlapping sources on the 16-point MSIDS model, can increase chronic symptoms [218]. These *Borrelia* aggregates were recently shown to be a major factor in increasing immunogenicity and antigenicity in late-stage Lyme disease patient’s sera, reaching statistical significance in *Borrelia burgdorferi sensu stricto* and also in *B. garinii* [219], the European agent of neurological Lyme disease. Italian researchers have implicated the biofilm-like aggregates of *Borrelia* as playing an important role in neuroborreliosis [220], where they can lead to antibiotic resistance and the reoccurrence of Lyme disease, allowing *Borrelia* spp. to resist adverse environmental conditions. Biofilm-like aggregates for *Bb* have been found in vitro in aged cultures but also the midgut of ticks [221,222], which could explain in part why some patients with early Lyme disease do not adequately respond to antibiotics. Biofilm-like aggregates create antibiotic resistance/tolerance [223], and a randomized controlled study using biofilm agents with antibiotics such as doxycycline in early Lyme disease is warranted based on the worldwide spread of the disease, with increasing numbers of affected individuals, and the fact that up to 30% of patients treated for early LD can go on to develop CLD/PTLDS [224].

Pulsing antibiotics has been found to effectively lower the biofilm forms of different bacteria since biofilms create antibiotic tolerance, leading to chronic infections and recalcitrant symptoms [225]. Common biofilm infections include periodontitis, implant-associated infections, endocarditis, cystic fibrosis pneumonia, chronic otitis media, chronic sinusitis, recurrent urinary tract infections, and chronic wound infections [226]. In the case of *Staphylococcus aureus*, a frequent pathogen in wound infections, the continuous administration of oxacillin does not decrease the level of persisters in culture, but when pulsed with properly timed breaks, it decreases the surviving population [225]. In that study, the length of the periodic break in between antibiotics impacted efficacy, with an optimal length that sensitized the biofilm to repeat treatment without allowing resistance. The optimal length of time in between pulses to rapidly eradicate persisters has been recently addressed using a methodology where the bactericidal effectiveness of periodic pulse dosing depended primarily on the ratio of durations of the corresponding on and off parts of the pulse [227]. The methodology relies on explicit formulas that make use of easily obtainable data from time-growth and time-kill experiments with a bacterial population exposed to antibiotics [227] and could be conducted for both *Borrelia burgdorferi* and *Bartonella henselae* since both have biofilm/persister forms in culture and in vitro [82,83,84,153,158,160,228,229].

Dr Kim Lewis previously demonstrated that four pulse doses of ceftriaxone killed persisters, eradicating all live bacteria in the culture, whereas *B. burgdorferi* persisters were otherwise capable of surviving at very high concentrations of continuous antibiotics, in doses exceeding what is clinically achievable [81]. Other authors have found that ceftriaxone pulse dosing fails to eradicate biofilm-like microcolony *B. burgdorferi* persisters [230] and that the use of a persister medication like IV Daptomycin combined with doxycycline and cefuroxime was more effective [230]. The conclusion of that study was that pulse dosing depends on the specific drugs used, with bactericidal drugs being superior for pulse dosing than static or persister drugs, although drug combinations with persister drugs were more effective at killing the more resistant microcolony form of persisters than pulse dosing [230]. In our three patients, dapsone combination therapy used four different persister drugs in combination: dapsone, rifampin, pyrazinamide, and methylene blue [77,79,231,232], along with doxycycline and azithromycin, which are generally considered bacteriostatic antibiotics, although they can occasionally exhibit bactericidal effects at higher concentrations [233]. In our three patients, after an initial 8-week course of DDDCT, pulsing higher dose dapsone, i.e., 2-week pulses of HDDCT, especially using 6-day pulses, was effective in putting all three patients in remission. Patient one did not require multiple pulses of dapsone, likely because the biofilm/persister forms of *Borrelia* and *Bartonella* had previously been adequately lowered using months of lower dose dapsone years prior, up to 100 mg per day. She received one 6-day pulse of HDDCT and is now over one year in remission. Impressively, she did not have any relapse of Lyme disease or *Bartonella* symptoms after recently taking a 5-day course of moxifloxacin for resistant bronchitis, as opposed to patient number 20 in our initial publication, who relapsed one year later with a *Bartonella* relapse and emergence of *Bartonella* striae after being treated for cellulitis. For patients 2 and 3, six pulses were needed over several years to achieve a more enduring remission, but in both cases, it was not until one or two 6-day pulses of HDDCT were completed that both had a more complete remission of symptoms. Patient 2 received four 4-day pulses of HDDCT followed by two 6-day pulses of HDDCT. Patient 3 received five 4-day pulses of HDDCT, but it was not until a final sixth pulse, using six days of HDDCT, that she was able to remain in long-term remission for over one year. Using multiple persister drugs in combination, along with biofilm agents and intermittent pulsing, was an effective strategy in all three patients, and it clinically appeared as if we were lowering the biofilm/persister aggregates over time, a primary culprit in relapsing and remitting symptoms.

## 5. Conclusions

Lyme disease and its associated co-infections, including *Babesia* and *Bartonella*, are increasingly being reported in the medical literature as potential overlapping factors leading to chronic resistant symptomatology [14,38,39,63,73,234,235], where combination antibiotics effectively relieve tick-borne infection (TBI) symptoms, with good patient tolerance [236]. Our three patients with *Borrelia*, *Bartonella*, and/or *Babesia,* described in these case reports, all had excellent outcomes with short-term combination antibiotics, using pulsed persister drug regimens with biofilm agents. *Bartonella* specifically required one or more 6-day pulses of HDDCT in order for patients to achieve long-term remission, although the number of pulses of HDDCT needed to cure *Bartonella* will require further investigation. Multiple overlapping sources of inflammation on the 16-point MSIDS model (infections, toxins, leaky gut and/or food sensitivities with mast cell activation, microbiome abnormalities, nutritional deficiencies, and/or insomnia) with downstream effects of inflammation (mitochondrial dysfunction, POTS/dysautonomia, autoimmunity, hormonal dysregulation, pain syndromes, liver and neurological dysfunction, and/or neuropsychiatric manifestations) were also found to be playing a significant role in driving underlying chronic symptomatology [30,36,37]. Infection, immune dysfunction, and inflammation, i.e., the three ‘I’s’, and all 16 factors on the MSIDS model, have now been reported in the medical literature as being potential factors that increase the symptoms of CLD/PTLDS as well the symptoms of long COVID [40,41,42,43,44,45,46,47,48,49,50,51,52,53,54,55,56,57,58,59]. A paradigm shift from a one-cause/one-disease model to a multifactorial model in chronic disease is warranted based on the increasing numbers of individuals affected by both illnesses.

The role of co-infections in CLD/PTLDS is the topic of an ongoing debate, especially *Bartonella*, since its transmission via ticks has only been proven in one European study of *Ixodes ricinus* ticks to date [237], although the 2018 HHS Tick-borne Disease Working Group report to Congress highlighted the importance of *Bartonella* species complicating tick-borne infections in humans [190]. All three of our Lyme disease patients had overlapping co-infections complicating their clinical course, especially *Bartonella*, with associated neuropsychiatric symptoms. These included moderate to severe cognitive difficulties, neuropathy, double vision and/or visual loss, dizziness, tremors, twitching, tinnitus, anxiety, depression, OCD, psychosis, and paranoid ideation, although headaches, fatigue, and/or malaise are the only two tick-borne disease symptoms fully recognized by public health officials [238]. Among 148 patients who were recently surveyed as having Lyme disease or another TBD, bartonellosis, Lyme disease, and babesiosis were the three most-reported TBDs, where the authors noted a disconnect between the scholarly literature regarding the psychiatric manifestations of LD and the lack of inclusion of psychiatric symptoms by the CDC [236]. This has particular importance from a public health perspective, considering that Lyme disease has been associated with suicidal ideation [239], where individuals with Lyme borreliosis have higher rates of any mental or affective disorders, suicide attempts, and death by suicide compared to those without Lyme borreliosis [240]. Similarly, bartonellosis has recently been reported to have a high association with neuropsychiatric symptoms, especially in those presenting with *Bartonella* striae [163,164,241], which were present in two out of three of our patients. The treatment of Lyme disease and its associated co-infections, including *Bartonella*, using pulsed dapsone combination therapy relieved these resistant neuropsychiatric symptoms that had not been previously controlled with psychiatric medications. Overlapping psychosocial stressors also needed to be addressed in helping to heal from these TBDs, whether from abuse and trauma or the COVID-19 pandemic. The mental health consequences of the COVID-19 pandemic has now been associated with severe mental health issues in the global population, including relatively high rates of anxiety, depression, PTSD, and psychological distress in the population [242], where Lyme disease may be a mitigating factor that increases symptoms [24]. Climate change is increasing the rates of Lyme disease and other vector-borne diseases, including viral infections [243], and *Bartonella* [244], whether from tick bites or other arthropod vectors, including fleas, spiders, mites, red ants, keds, or lice, requires a One Health approach to mitigate the rise in vector-borne chronic illnesses, improve physical and mental health, and lower rising health care costs [245,246,247].

Future studies need to be conducted on persister drugs like dapsone, using pulsed antibiotics with biofilm agents, and to further elucidate the role of associated co-infections like *Bartonella* with MSIDS variables in those suffering from CLD/PTLDS. Our report represents one of the first successful treatments of a pediatric patient with CLD using a persister drug regimen, i.e., dapsone combination therapy. Future studies should be conducted in both the adult and pediatric populations since pediatric Lyme borreliosis (LB) represents a substantial proportion of affected individuals across countries in Europe and North America [248]. The four prior National Institutes of Health (NIH) randomized trials on treatments for CLD, conducted over 15 years ago, did not evaluate pulsing, combination persister drug regimens, and/or MSIDS variables in the treatment of those suffering from chronic Lyme symptoms, although two of the four U.S. treatment trials demonstrated the efficacy of IV ceftriaxone on primary and/or secondary outcome measures [249]. A multicenter, placebo-controlled, randomized trial using dapsone combination therapy to evaluate the role of co-infections including *Babesia* and *Bartonella* and screen for overlapping MSIDS variables is the next logical step to help end the decades-long scientific debate over the etiology and treatment of CLD/PTLDS. Based on the number of increasing cases of TBDs and vector-borne infections due to climate change [250], resulting in increased health inequities and associated patient suffering, disability, and rising healthcare costs [251], we urge Lyme groups and healthcare authorities to come together now to help solve this urgent global healthcare crisis [252].

## Figures and Tables

**Table 1 microorganisms-12-00909-t001:** Summary of Co-infection Status, MSIDS Variables, and DDDCT and HDDCT Pulses Leading to Clinical Remission.

Patient	Co-Infection Status	MSIDS Variables Affecting Treatment Outcome	Number of DDDCT Treatments	Number of 4-Day Pulses of HDDCT	Number of 6–7-Day Pulses of HDDCT	Time in Remission
Case 1	*Babesia*, *Bartonella*, COVID exposure, and *Bartonella* FISH+	Metabolic syndrome/hypoglycemia Alpha-1 antitrypsin deficiency Chronic variable immune deficiency (CVID) Mast cell activation syndrome (MCAS) Phase 2 adrenal dysfunction Mycotoxins (aflatoxins, ochratoxins, trichothecenes, and gliotoxins) Heavy metal exposure (lead) Postural orthostatic tachycardia syndrome (POTS)/dysautonomia	1	0	1	Greater than 1 year
Case 2	*Babesia*, *Bartonella*, COVID exposure, *Bartonella* FISH+, and *Bartonella* striae +	B12 and Vitamin D deficiency Low glutathione levels Phase 3 adrenal insufficiency Mycotoxin exposure (trichothecenes, gliotoxins, and zearalenone) Borderline elevated levels of heavy metals (aluminum) Mild to moderate POTS/dysautonomia with vagal dysfunction and gastroparesis Intermittent supraventricular tachycardia (SVT) Reactive hypoglycemia Candida and leaky gut with multiple food sensitivities Possible mast cell disorder Severe anxiety, obsessive–compulsive disorder (OCD), paranoia, and psychosis	1	4	2	2 months after the last HDDCT pulse
Case 3	*Anaplasma*, *Babesia*, *Bartonella*, *Bartonella* striae+, relapsing fever *Borrelia*, recurrent HSV-1, HSV-2, and HPV, and COVID × 6	Vitamin B and Vitamin D deficiency Mineral deficiency (iodine, iron, and zinc) Low glutathione levels Microbiome abnormalities Phase 3 adrenal dysfunction Postural orthostatic tachycardia syndrome (POTS) Probable mast cell activation (MCAS) Elevated serum heavy metals (mercury and aluminum) Mycotoxin exposure (ochratoxins, aflatoxins, trichothecenes, gliotoxins, and zearalenone) Reactive hypoglycemia, insulin resistance with polycystic ovarian syndrome (PCOS) Candidiasis HLA DR 4 positivity with elevated autoimmune markers Depression, anxiety, premenstrual dysphoric disorder (PMDD), and Post-traumatic stress disorder (PTSD) Chronic insomnia and restless leg syndrome Possible mitochondrial dysfunction	1	5	1	Greater than 1 year

Abbreviations: Multiple Systemic Infectious Disease Syndrome (MSIDS); Double-Dose Dapsone Combination Therapy (DDDCT); and High-Dose Dapsone Combination Therapy (HDDCT).

## Data Availability

Data are contained within the article.

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
