# Peer review of "Combining Double-Dose and High-Dose Pulsed Dapsone Combination Therapy for Chronic Lyme Disease/Post-Treatment Lyme Disease Syndrome and Co-Infections, Including Bartonella: A Report of 3 Cases and a Literature Review"

_microorganisms, 2024, doi:10.3390/microorganisms12050909_

Round 1
Reviewer 1 Report
Comments and Suggestions for Authors
The manuscript entitled "Treatment of Chronic Lyme Disease/ Post Treatment Lyme Disease Syndrome and Associated Co-infections including Bartonella With High Dose Pulsed Dapsone Combination Therapy: A report of 3 cases and Literature Review" is a well-structured paper that addresses 3 case studies (Lyme disease) with co-infections of other hemobacteria (Bartonella) and hemoparasites (Babesia) in unsuccessfully treated individuals. These female individuals had records of COVID-19 infection and subsequent complications.
Some comments:
Title: In developing the case studies, the authors mention combining a combination therapy scheme with a double dose of dapsone (DDDCT) and combined treatment with a high dose of dapsone (HDDCT). I suggest modifying the title to combined therapies using dapsone.
Line 29: Keywords, I consider that the word "Florescent In Situ Hybridization (FISH)" should not be included since, although in the case studies, the authors mention that they performed this diagnostic test, they do not mention a methodology for performing it, considering that other diagnostic tests (hemogram and serum biochemistry) were performed to evaluate the general health of the patients. In case of placing it, correct it to "fluorescent."
- Although the authors showed a section on how to minimize the possible side effects of these therapeutic protocols (3. Minimizing Potential Side Effects of Dapsone Combination Therapy/Prescribing Information) and in each clinical case, it is mentioned which drugs and food supplements were administered to each case, I consider it convenient to say (although it is a novel therapeutic scheme in these patients) that depending on the case it is necessary to evaluate the cost-benefit of carrying out this treatment.
- Lines 2134- 2139: "Regarding dapsone's efficacy in Lyme disease, dapsone alone, and in combination with rifampin, and tetracycline and a macrolide (azithromycin) is effective in decreasing the biofilm/persister aggregates of Borrelia burgdorferi in culture. Adding another antibiotic to dapsone (doxycycline, rifampin, and azithromycin) increased the antibiotic combination's efficacy and ability to lower the bacteria's biofilm load". This manuscript mentions three case studies where dapsone is combined with other antibiotics/drugs, such as rifampicin, tetracycline, doxycycline, or macrolides. However, it mentions evidence that dapsone in combination has higher efficacy. If there is evidence that using it individually effectively reduces Borrelia burgdorferi aggregates under laboratory conditions, could it be more convenient to use therapeutic schemes with dapsone alone? No [double dose dapsone (DDDCT) and high dose dapsone combination therapy (HDDCT)], since many drugs or dietary supplements are used to prevent possible side effects. What about patients with renal or hepatic impairment and problems metabolizing medications?
- All three case studies are female patients with favorable outcomes after treatment. However, in the 3 studies mentioned, during the menstrual cycle, they presented signs of Lyme disease such as joint pain, dizziness, mental confusion, fatigue, and joint stiffness, even in one case, problems with insomnia. To what is this finding attributed? Did the patients, after finishing their treatment and being on their period, still present Lyme symptoms, or did these symptoms diminish? And finally, could the authors mention recommendations to reduce these symptoms during the menstrual period of these patients?
Reviewer 2 Report
Comments and Suggestions for Authors
A significant percentage of patients with Lyme Disease (LD) may develop Post Treatment Lyme Disease Syndrome (PTLDS), a chronic disabling disease, with persistent neurocognitive difficulties, which significantly affect their quality of life.
Multiple systemic infectious disease syndrome (MSIDS), such as in the case of Long COVID, is also considered, also as both these diseases share similar pathogenetic mechanisms. There is also to consider the forms of HTBRF, STBRF and Bartonella and Babesia infections, which can increase the severity of LD. Borrelia burgdorferi sensu lato (Bbsl) and Bartonella spp. can become persistent bacteria and the logarithmic phase can be followed by the stationary phase.
Furthermore, an obstacle to the response of antibiotics is the formation of biofilms, which can be overcome with drugs such as Rifmpicin and Dapsone. It is interesting to note that these two drugs are used in the treatment of Leprosy, and in this disease, Rifampicin is administered in a pulsed manner, while in this scheme it is Dapsone that is pulsed. For this purpose, three patients with a history of Bbsl infection, and Bartonella and/or Babesia co-infection were treated sequentially using two treatment schemes: Pulsed High Dose Dapsone 12 Combination Therapy (HDDCT) and Double dose dapsone combination therapy 11 (DDDCT) , which were used sequentially.
The three patients were a 12-year-old white girl, a 21-year-old white woman, and a 35-year-old white woman. The treatment was carried out using HDDCT and DDDCT sequentially, with personalized schemes.
Before treatment the patients were tested for G-6-P-D, which was normal.
During and at the end of the treatment, the parameters linked to the administration of Dapsone (methemoglobin) and organ functions were monitored. The treatment led to remission of the clinical picture, which is maintained months/years later.
Based on the increasing number of cases of vector-borne infections and associated patient suffering and disability, the authors suggest that Lyme groups and health authorities come together to help resolve this health emergency.
The work is written in great detail, and can be accepted in this form.
Reviewer 3 Report
Comments and Suggestions for Authors
Dear Authors
I had the honour of reviewing the manuscript entitled "Treatment of Chronic Lyme Disease/ Post Treatment Lyme Disease Syndrome and Associated Co-infections including Bartonella With High Dose Pulsed Dapsone Combination Therapy: A Report of 3 Cases and Literature Review".
In my opinion, it would be better to try to describe the symptoms of the three cases in a different way, perhaps by adding summary tables. The risk of leaving such a long text is that the reader loses focus on the focal point of the study. The ability to summarise is a fundamental characteristic in publishing in scientific journals precisely so that those who read our work are interested from beginning to end.
The manuscript needs less revision to summarise the study.
